# Tetraspanin-8 sequesters syntaxin-2 to control biphasic release propensity of mucin granules

José Wojnacki[1], Agustin Leonardo Lujan[1], Nathalie Brouwers[1], Carla Aranda-Vallejo [1,2], Gonzalo Bigliani[1], Maria Pena Rodriguez[1], Ombretta Foresti [1] & Vivek Malhotra [1,2,3] ✉

Agonist-mediated stimulated pathway of mucin and insulin release are biphasic in which rapid fusion of pre-docked granules is followed by slow docking and fusion of granules from the reserve pool. Here, based on a cell-culture system, we show that plasma membrane-located tetraspanin-8 sequesters syntaxin-2 to control mucin release. Tetraspanin-8 affects fusion of granules during the second phase of stimulated mucin release. The tetraspanin-8/syntaxin-2 complex does not contain VAMP-8, which functions with syntaxin-2 to mediate granule fusion. We suggest that by sequestering syntaxin-2, tetraspanin-8 prevents docking of granules from the reserve pool. In the absence of tetraspanin-8, more syntaxin-2 is available for docking and fusion of granules and thus doubles the quantities of mucins secreted. This principle also applies to insulin release and we suggest a cell type specific Tetraspanin/Syntaxin combination is a general mechanism regulating the fusion of dense core granules.

A key issue in cell and tissue function is how cells secrete the right quantities of proteins. This becomes especially crucial for proteins such as neurotransmitters, insulin, and mucins. Cells have developed systems that allow these kinds of proteins to be secreted at a slow but constant rate and also, an agonist-dependent stimulated release. When an external agonist binds to specific receptors, it causes an increase in intracellular calcium that triggers the fusion of large granules containing insulin or mucins[1-3] or in neurons, small synaptic vesicles that store neurotransmitters[4].

Humans express 5 gel-forming mucins that are secreted by specialized goblet cells[5-9]. Secreted mucins mix with other extracellular components to constitute the mucus, which acts as a lubricant and a protective barrier to the underlying epithelium[10-12]. But, as noted in human pathologies of the airways and the digestive system[13-15], too much or too little extracellular mucins may be damaging, which begs the question, how cells control the quantity and, for that matter, quality[16] of mucins secreted.

Post sorting and packing at the Golgi into micro meter size granules, mucins are condensed. These condensed mucin-filled granules fuse to the plasma membrane at a constant low rate (basal secretion) and, if necessary, an external agonist triggers a massive release of mucins in a short period of time (stimulated secretion)[17-19]. The stimulated mucin secretion is biphasic. Within seconds after agonist stimulation the rate of mucin secretion increases more than a thousand times compared to baseline secretion. The short burst of secretion (less than 1 min) is followed by a longer, slower-rate phase. The second phase lasts several minutes during which, the secretion rate is 47 times slower compared to the peak rate but still 38 times higher compared to baseline secretion[17]. It is generally assumed that the rapid release involves fusion of pre-docked vesicles, whereas the slower-sustained release involves fusion of vesicles/granules in reserve. The availability of specific sites, to which vesicles can dock, might be the major bottleneck controlling fusion[20], but how this is regulated remains unexplored. The highly conserved, SNARE proteins

---

[1]Centre for Genomic Regulation (CRG), The Barcelona Institute for Science and Technology, Barcelona, Spain. [2]Universitat Pompeu Fabra (UPF), Barcelona, Spain. [3]ICREA, Barcelona, Spain. ✉e-mail: vivek.malhotra@crg.eu

are essential for basal and stimulated granule fusion[4,18,21,22]. Mucin granules bearing the R-SNARE, Vesicle-associated membrane protein 8 (VAMP-8), fuse to the plasma membrane to release their contents[21,23]. During exocytosis, VAMP proteins interact with syntaxins (Q-SNAREs) in trans, to generate the necessary force for membrane fusion[4,24,25]. Cells from the human airways express syntaxins 1, 2, 3 and 4[26–28], but the identity of the syntaxin required for mucin secretion in intact cells remains unknown. Syntaxins are also required to dock secretory granules to the plasma membrane[29–31]. It has been suggested that a release site (vesicle docking site) can exist in three different states: (i) empty and accessible for a vesicle (ii) occupied, ready for its vesicle to exocytose (iii) empty and refractory (not accessible for a vesicle)[20]. Cells could control the availability of Q-SNAREs at the plasma membrane to increase or decrease the number of accessible docking sites and the quantity of released material during secretion.

We report here that syntaxin-2 (Stx2) is present at the apical plasma membrane of mucin-secreting cells and required for mucin secretion. A more surprising finding is that tetraspanin-8 (Tspan-8), which also localizes to the plasma membrane, binds syntaxin-2 (and syntaxin-3) and controls its availability for SNARE-dependent mucin secretion. Stx2 is either bound to VAMP-8 or to Tspan-8 and when in complex with Tspan-8, Stx2 is unavailable to engage to VAMP-8. Loss of Tspan-8 increases mucin release by the stimulated pathway, specifically during the second phase of release. In the absence of Tspan-8, the biphasic release of mucins is deregulated. Tetraspanin-8 thus emerges as a component that controls the release propensity of mucin granules by the external agonist stimulated pathway.

## Results

### Upregulation of tetraspanin-8 in mucin secreting goblet cells

Upon differentiation, goblet cells upregulate mucin (MUC) genes and we asked a simple question. Are there genes, other than the mucins, that are upregulated during this transition? A bioinformatics analysis of a single-cell RNA sequencing (scRNAseq) profiling database of healthy human airways[32] was thus performed. We compared the scRNAseq profiles of 17,712 secretory cells with 24,138 basal cells from four regions of the human airways: nasal, proximal, intermediate and distal (Supplementary Fig. 1a). For each of these regions we calculated the fold change in gene expression between secretory and basal cells. Most genes were repressed in differentiated secretory cells, which is shown as puncta below the horizontal dashed line in the dot plot (Fig. 1a and Supplementary 1b). Genes above the horizontal line were upregulated in secretory cells. As expected, MUC5AC and MUC5B are highly upregulated, but of interest is the finding that TSPAN8 was also upregulated (Fig. 1a and Supplementary 1b). The function of TSPAN8 and its enrichment in goblet cells has not been described thus far. TSPAN8 expression in secretory cells from all areas of the airways was on average 42% higher than in basal cells and comparable to two secreted mucins, MUC5AC and MUC5B (Fig. 1b). Less than 10% of basal cells expressed TSPAN8, MUC5AC or MUC5B while the same genes were expressed in over 75% of differentiated secretory cells. The proportion of basal, parabasal, multi ciliated and secretory cells expressing ATG5 and GAPDH was similar in all cell types (Fig. 1c). We found that at least 60% of secretory cells co-expressed TSPAN8, MUC5B and MUC5AC (Fig. 1d).

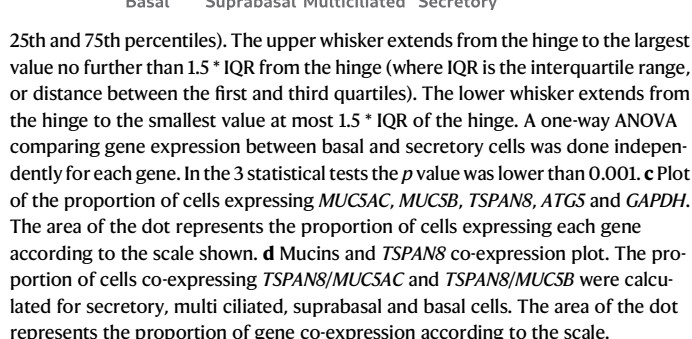

**Fig. 1 | Tetraspanin-8 is upregulated in mucin-secreting cells from the healthy human airways. a** Dot plot showing fold-change in gene expression between basal and mucin-secreting cells in the y-axis and the proportion of mucin-secreting cells expressing each gene in the x-axis. Each dot represents one gene of the distal airways. 18,417 genes were analyzed. Larger dots represent MUC5AC, MUC5B and TSPAN8 genes. Dots are plotted with a 50% transparency value for better visibility in areas of the graph with high dots density. **b** Box and dot plots of the expression levels of MUC5AC, MUC5B and TSPAN8. Cells from the nasal, proximal, intermediate and distal regions of the airways were pooled together. Each dot represents gene expression in a single cell. 41,850 cells were analyzed. Cells in which no expression was detected are lined at the bottom of the y-axis. The boxes of the box plots were calculated excluding cells with no detectable expression. The lower and upper hinges of the box plots correspond to the first and third quartiles respectively (the 25th and 75th percentiles). The upper whisker extends from the hinge to the largest value no further than 1.5 * IQR from the hinge (where IQR is the interquartile range, or distance between the first and third quartiles). The lower whisker extends from the hinge to the smallest value at most 1.5 * IQR of the hinge. A one-way ANOVA comparing gene expression between basal and secretory cells was done independently for each gene. In the 3 statistical tests the p value was lower than 0.001. **c** Plot of the proportion of cells expressing MUC5AC, MUC5B, TSPAN8, ATG5 and GAPDH. The area of the dot represents the proportion of cells expressing each gene according to the scale shown. **d** Mucins and TSPAN8 co-expression plot. The proportion of cells co-expressing TSPAN8/MUC5AC and TSPAN8/MUC5B were calculated for secretory, multi ciliated, suprabasal and basal cells. The area of the dot represents the proportion of gene co-expression according to the scale.

To test the role of *TSPAN8* in mucin secretion, we used the human-derived and mucin-secreting cell line HT29-N2[33,34]. A western blot analysis showed that HT29-N2 cells express *TSPAN8* and that its expression is elevated in differentiated cells (Supplementary Fig. 1c and d).

## Tetraspanin-8 depletion causes mucin hyper-secretion

The large size of gel-forming mucins and the complex conditions to grow goblet cells in culture has limited the use of standard molecular biology approaches and imaging systems necessary to unravel the mechanism of mucin secretion. We have taken a reductionist approach of using a cancer-derived, mucin secreting cell line that is easy to manipulate genetically to help identify genes required for mucin secretion and to understand their mechanism of action. With this in mind, we first created a cell line that expresses a tagged gel-forming mucin under endogenous genetic regulation. We tagged the c-terminal tail of mucin-5AC by inserting the sequence of the super-folded Green Fluorescent Protein (sfGFP)[35] to the *MUC5AC* locus by CRISPR/Cas9 genome editing. The GFP-tagged mucin-5AC-secreting cell line differentiated as wild type cells shown by the gradual increase in the levels of mucin-5AC during differentiation (Supplementary Fig. 2a and b). Basal and ATP-stimulated mucin-5AC secretion were not affected in mucin-5AC·GFP-tagged cells (Supplementary Fig. 2c and d; Supplementary Video 1). The total mucin-5AC content was not altered by its tagging with sfGFP (Supplementary Fig. 2e). In differentiated cells, mucin-5AC·GFP localized to distinct granules that showed high co-localization with immunolabelled mucin-5AC (Supplementary Fig. 2f; arrows). We also detected a minor population of GFP-positive granules, close to the perinuclear region, that were not recognized by the anti mucin-5AC antibody, suggesting that these could be immature mucin, without a mature epitope (Supplementary Fig. 2f; arrowheads). Mucin-5AC·GFP did not co-localize with markers of the Golgi apparatus or with the lysosomal marker Lamp1 (Supplementary Fig. 2g–i) suggesting that most GFP-positive granules are post-Golgi and that mucin-5AC·GFP is not en route for degradation. These data show that GFP-tagged and untagged mucin-5AC behave identically, and we could use this genetically-engineered cell line to visualize mucin-5AC localization and secretion.

We knocked-out (KO) *TSPAN8* by CRISPR/Cas9 genome editing in the mucin-5AC GFP-tagged cell line. Complete loss of tetraspanin-8 mRNA and protein were confirmed by reverse transcription-polymerase chain reaction (RT-PCR) and western blot, respectively (Supplementary Fig. 3a and b). A secretion assay showed that basal mucin-5AC·GFP secretion was unaltered, while ATP-stimulated secretion was increased by more than 50% in two independent *TSPAN8* KO cell lines (Fig. 2a, b). Western blot analysis of the cell lysates showed the absence of Tspan-8 (Fig. 2c). Equal amount of cells in each condition was confirmed by western blot analysis of beta-tubulin in the cell lysates (Supplementary Fig. 3c and d). Mucin-5AC·GFP expression was not changed by *TSPAN8* KO (Fig. 2d, e). Tspan-8 loss and equal amount of cells were confirmed by western blot analysis of the lysates (Fig. 2f). Beta-tubulin in the secreted fractions was never greater than 3% of the total amount, statistically equal between WT and *TSPAN8* KO cells, and unaffected by ATP stimulation (Supplementary Fig. 3e and f). The increased mucin secretion upon *TSPAN8* loss is therefore not due to altered gene expression or cell lysis.

To determine how Tspan-8 affected stimulated secretion, we imaged mucin-5AC·GFP in live WT and *TSPAN8* KO (clone14) cells by spinning disk microscopy during ATP-stimulated mucin secretion. WT and *TSPAN8* KO cells responded similarly during the fast-rate secretory phase by releasing 20% of mucin-5AC·GFP granules within the first minute after ATP addition (Fig. 2g–i; Supplementary Videos 2 and 3). During the slow-rate phase of stimulated secretion, WT cells released 0.33% of granules each minute (statistical linear model: percentage of granules ~ time; $p < 0.001$; $R^2$: 0.907). At this rate, during the 24 min of the slow phase of stimulated secretion, WT cells had secreted 7.9%

more of the total amount of granules (Fig. 2g, h; Supplementary Videos 2 and 3). *TSPAN8* KO cells on the other hand, more than doubled the rate of release to 0.73% (statistical linear model: percentage of granules ~ time; $p < 0.001$; $R^2$: 0.951). During the 24 min of the slow-rate phase of stimulated secretion, *TSPAN8* KO cells had secreted 17.5% of the total amount of granules (Fig. 2g, i; Supplementary Videos 2 and 3).

## Tetraspanin-8 is localized at the plasma membrane

Transient transfection of Tspan-8·RFP and immunofluorescence-based visualization of the plasma membrane-localized sodium/potassium-transporting ATPase alpha-1 subunit (Na$^+$/K$^+$-ATPase α1) followed by an analysis of the Pearson Correlation Coefficient (PCC) showed high co-localization between the two proteins (Fig. 3a, b; Supplementary Video 4). To exclude the possibility that the localization of Tspan-8 could be affected by over expression or transfection, we tagged the c-terminus of Tspan-8 by inserting the sfGFP sequence in the *TSPAN8* locus by CRISPR/Cas9 genome editing (Supplementary Fig. 4a). A mucin secretion assay and dot blot analysis showed that the c-terminus tagging of Tspan-8 did not affect mucin production (Supplementary Fig. 4b) or secretion (Supplementary Fig. 4c and d). Live cell imaging of Tspan-8·GFP showed co-localization with the lipophilic plasma membrane-marker CellBrite® (Fig. 3c; arrows; Supplementary Video 5). Immunolabeling of Tspan-8 in WT cells confirmed its localization at the plasma membrane and absence from mucin granules (Supplementary Fig. 4e; arrows). An intracellular pool of Tspan-8 co-localized with the recycling endosomes marker Rab11A (Supplementary Fig. 4f; arrowheads), which fits well with the known recycling behaviour of plasma membrane proteins[36,37].

We obtained a plasma membrane enriched fraction (see materials and methods) and western blotted with antibodies to Na$^+$/K$^+$-ATPase α1 and Tspan-8. These data further confirmed the presence of Tspan-8 along with Na$^+$/K$^+$-ATPase α1 at the plasma membrane (Fig. 3d).

## Tetraspanin-8 interacts with the Q-SNAREs syntaxin-2 and syntaxin-3

The localization of Tspan-8 and its effect on mucin secretion suggested that it likely functions in the terminal stages of mucin granules docking/fusion to the plasma membrane. We immunoprecipitated Tspan-8·GFP from differentiated mucin-secreting cells and determined by western blot whether plasma membrane-localized syntaxins 1, 2, 3 or 4 were present in the immunoprecipitated samples. Syntaxins 2 and 3 were found to co-immunoprecipitate with Tspan-8·GFP (Fig. 4a). Syntaxins bind R-SNAREs in events leading to membrane fusion[25,38]. VAMP-8 is the R-SNARE that localizes to mucin granules and mediates their fusion to the plasma membrane[21,23]. Stx2 could therefore bind VAMP-8 for the fusion of mucin granules to the plasma membrane. We confirmed that Stx2 interacts with VAMP-8 by co-immuno-precipitation of GFP·Stx2 followed by western blot analysis of samples from WT HT29-N2 cells transiently transfected with GFP·Stx2 (Fig. 4b). Surprisingly, when we immunoprecipitated Tspan-8·GFP, we detected Stx2 in the protein complex but not VAMP-8 (Fig. 4c and Supplementary Fig. 5a and b). To confirm that VAMP-8 is not present in the Tspan-8/Stx2 protein complex, we immunoprecipitated VAMP-8 and asked whether Stx2 or Tspan-8 are contained in the immunoprecipitate. We detected syntaxin-2 in the precipitate, but Tspan-8 was not detected (Fig. 4d). Munc18-1 and 2 bind to syntaxins to regulate the formation of SNARE complexes[25,39] and are known for their involvement in mucin secretion[40]. By transiently co-transfecting Tspan-8·GFP and Stx2·FLAG and immunoprecipitating GFP, we reconfirmed Stx2 as a Tspan-8 interacting protein, but neither Munc18-1 (Fig. 4e) nor Munc18-2 (Fig. 4f) were found as part of the Tspan-8/Stx2 protein complex. These data nicely show that Stx2 is either in complex with Tspan-8 or VAMP-8, but not simultaneously with both (Fig. 4c, d and Supplementary Fig. 5a and b). To further confirm this observation, we transiently co-transfected cells with GFP·Stx2 and Tspan-8·RFP and immunoprecipitated GFP to

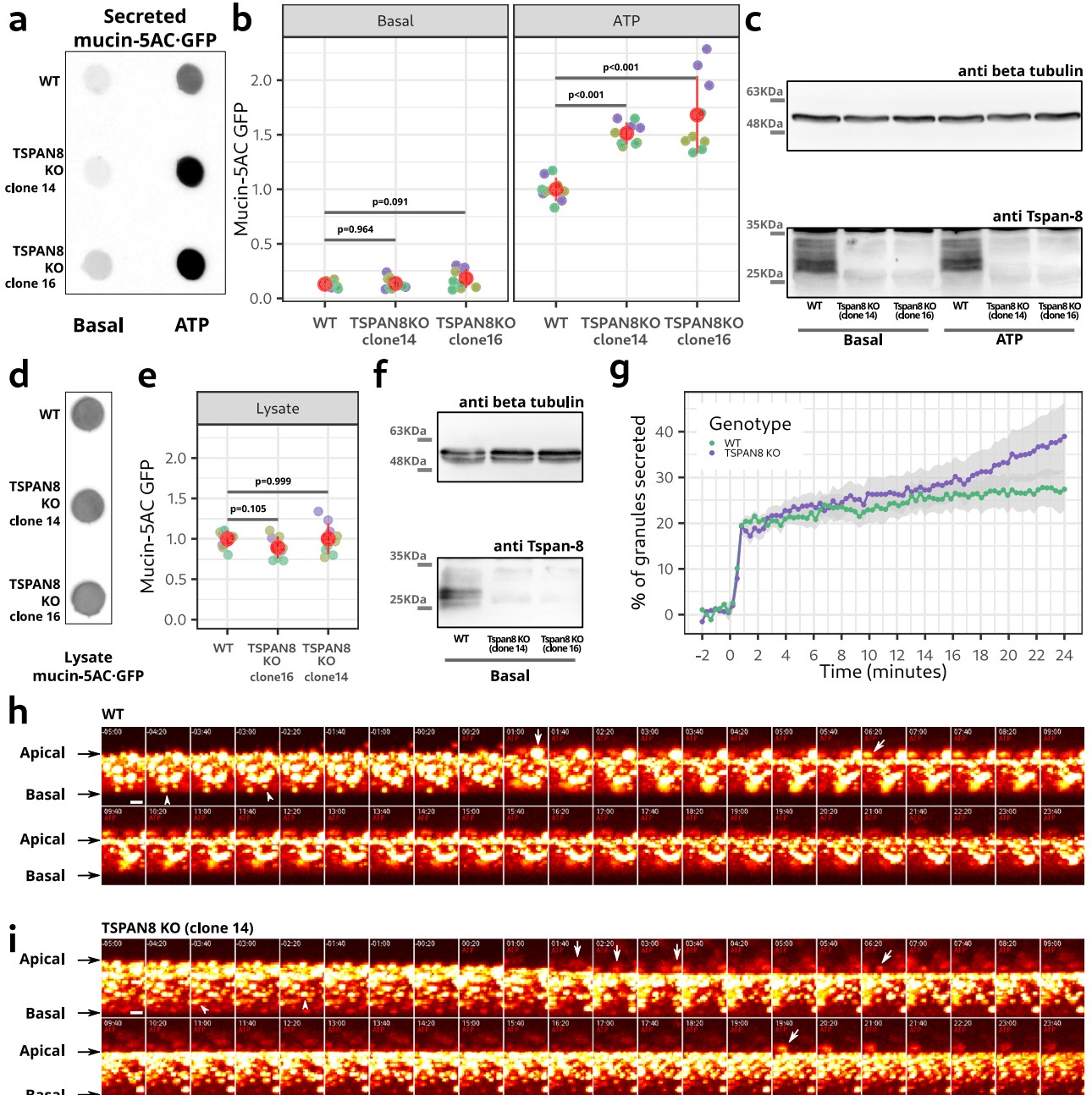

**Fig. 2 | *TSPAN8* KO increases mucin secretion. a** Representative dot blot showing mucin-5AC·GFP secreted by WT and *TSPAN8* KO cell lines. **b** Quantification of the amount of mucin-5AC·GFP in secreted fractions of WT and *TSPAN8* KO cell lines. Each dot represents the signal from one secretion assay. Grouped in different colors are secretion assays that were processed in parallel. $n = 9$. Red dots represent the mean +/− the standard deviation. Statistical analysis was performed independently for basal and ATP-stimulated conditions. The *p* values of planned orthogonal contrasts are shown. **c** Lysates of WT and *TSPAN8* KO cells of the secretion assay shown in **a** were processed for western blot analysis. **d** Representative dot blot of the total mucin-5AC·GFP in unstimulated WT and *TSPAN8* KO cell lines. **e** Quantification of the amount of mucin-5AC·GFP in the cell lysates of WT and *TSPAN8* KO cells. Each dot represents the signal from an independent sample. Grouped in different colors are samples that were processed in parallel. $n = 9$. Red dots represent the mean +/− the standard deviation. The *p* values of planned orthogonal contrasts are shown. **f** Lysates of WT and *TSPAN8* KO cells of the samples shown in **d** were processed for western blot analysis. **g** Quantification of secreted mucin granules in WT and *TSPAN8* KO cells during ATP stimulation. Each dot represents the average cumulative percentage of secreted granules. Values are expressed as proportional to time 0. The gray shadow represents the mean +/− the standard deviation. **h** Frames showing the lateral projection of a time-lapse of WT cells during ATP stimulation. Time is relative to the moment of ATP addition. Signal is from mucin5AC·GFP fluorescent emission and its intensity was color-coded in a pseudo-color look up table. Arrows point to mucin granules release. Arrowheads point to mucin granules inside cells before ATP stimulation. **i** Same as in **h**, but for *TSPAN8* KO cells. Source data for a to g are provided as a Source Data file.

precipitate the two protein complexes: Stx2/Tspan-8 and Stx2/VAMP-8. VAMP-8 and Tspan-8 in the precipitate confirmed the presence of both protein complexes (Supplementary Fig. 5c).

We then measured the interaction between syntaxin-2 and Tspan-8 by a procedure involving Förster Resonance Energy Transfer (FRET).

For FRET to occur, the donor and acceptor molecules can't be separated by more than 10 nanometres[41]. This distance between proteins can be considered direct interaction. We created two expression plasmids: neonGreen·Stx2 (energy donor) and Tspan-8·mScarlet (energy acceptor). These plasmids were used to create lentiviral

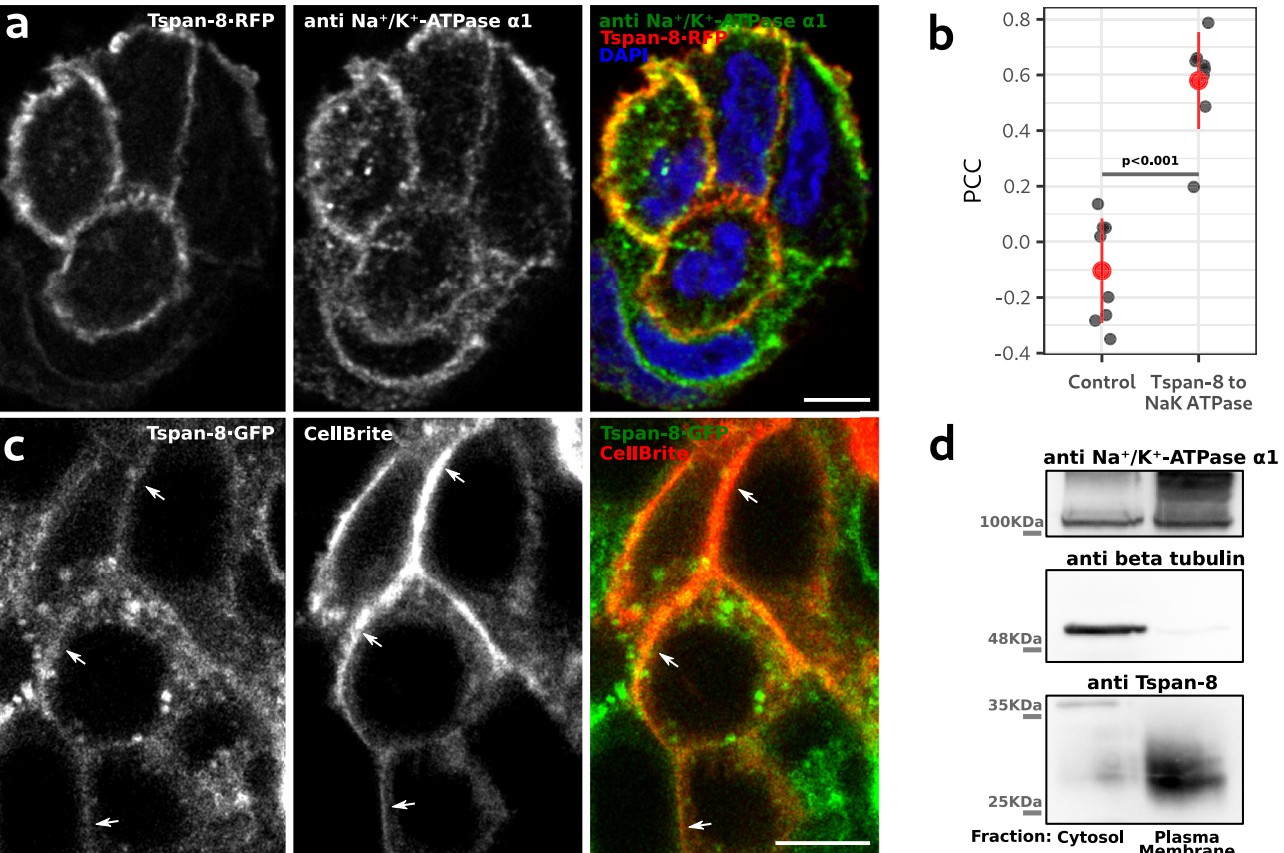

**Fig. 3 | Tetraspanin 8 is localized at the plasma membrane of mucin-secreting cells. a** Optical plane of a representative confocal image of mucin-secreting cells transfected with Tspan-8·RFP and immunolabelled for the Na⁺/K⁺-ATPase α1. Right image shows the merge of the two. Scale bar is 5 µm. **b** Quantification of the Pearson Correlation Coefficient (PCC) between Tspan-8·RFP and Na⁺/K⁺-ATPase α1. As control, Na⁺/K⁺-ATPase α1 images were rotated 90° right and the PCC was calculated. Each dot represents the PCC of one image with at least three cells. $n = 9$. The red dot is the mean value of the gray dots +/− the standard deviation. One-way ANOVA test $p < 0.001$. **c** Optical plane from a confocal image of live mucin-secreting cells expressing Tspan-8·GFP at endogenous levels. To visualize the plasma membrane, cells were incubated with the lipophilic membrane marker CellBrite®. Right image shows the merge of the two. Arrows point to the plasma membrane. Scale bar is 5 µm. Five images from two independent cell cultures showed similar results. **d** WT HT29-N2 cell lines were processed to obtain plasma membrane-enriched fractions. Purified samples were analyzed by western blot. As controls, top and middle panels show immunoblotting against Na⁺/K⁺-ATPase α1 and beta tubulin respectively. One of three independent experiment is shown. All replicates showed similar results. Source data for **b** and **d** are provided as a Source Data file.

particles. We coinfected HT29-N2 cells and selected only those cells co-expressing both proteins by Fluorescent Activated Cell Sorting (FACS) to generate stable cell lines. To measure FRET between neon-Green·Stx2 and Tspan-8·mScarlet, we used Acceptor Photobleaching for its simplicity and robustness. With this method we acquire an image of the energy donor, photobleach the acceptor and finally acquire a second image of the donor in the same conditions. If in the pre-photobleaching image there was FRET between Stx2 and Tspan-8, part of the energy used to excite the neonGreen of Stx2 would have been transferred to mScarlet to generate mScarlet emission (donor quenching). In the second image (post-photobleaching) there can not be any further transfer of energy as the acceptor is bleached, therefore, in the second donor image the intensity would be higher compared to the first image (unquenched donor image). Representative images of the energy donor pre- and post-photobleaching of the acceptor are shown in Fig. 4i. The increase in donor fluorescence is clearly seen in the post photobleaching image (Fig. 4i; arrows). Near complete pho-tobleaching of the acceptor is shown in Fig. 4g and quantified in Fig. 4h. A FRET map of the energy transfer efficiency shows that the interaction of Stx2 and Tspan-8 occurs at the plasma membrane (Fig. 4j; arrows). Quantification shows a 7 % FRET efficiency between Stx2 and Tspan-8 mScarlet and confirms their interaction (Fig. 4k). As a technical control of the FRET efficiency calculation, we acquired ima-ges pre and "post" photobleaching, but for which the laser was kept at

0% during the "bleaching" part of the protocol (Fig. 4k). This approach confirms the interaction between Tspan-8 and syntaxin-2 at the plasma membrane of mucin-secreting HT29-N2 cells.

## Syntaxin-2 is the plasma membrane Q-SNARE required for mucin secretion

We found that co-culture of HT29-N2 with Caco-2 cells[42] presents a highly polarized organization of the former and therefore a valuable approach to investigate polarized location of proteins like Tspan-8 and Stx2 and their involvement in polarized mucin secretion. We used this system to monitor the distribution of Stx2 between the basal and apical regions of HT29-N2 cells. Immunolabelling of Stx2 and Na⁺/K⁺-ATPase α1 in HT29-N2/Caco-2 co-cultures and quantification of the PCC in each optical slide, showed that Stx2 localized to the apical region of the plasma membrane (Fig. 5a, c; green line and dots). *TSPAN8* KO did not affect the localization of Stx2 (Fig. 5d, f; green line and dots). Specific analysis of Stx2 and mucin5AC·GFP distribution shows virtually no co-localization between the two in WT (Fig. 5b, c; violet line and dots) and in *TSPAN8* KO cells (Fig. 5e, f; violet line and dots). Consistent with the low PCC (Fig. 5c; violet lines and dots), Stx2 localized to the apical plasma membrane while the vast majority of mucin5AC localized in cytosolic granules in the apical region of cells (Fig. 5b). *TSPAN8* KO did not affect the location or distribution of Stx2 and mucin5-AC (Fig. 5e, f; violet lines and dots). The marginal increase

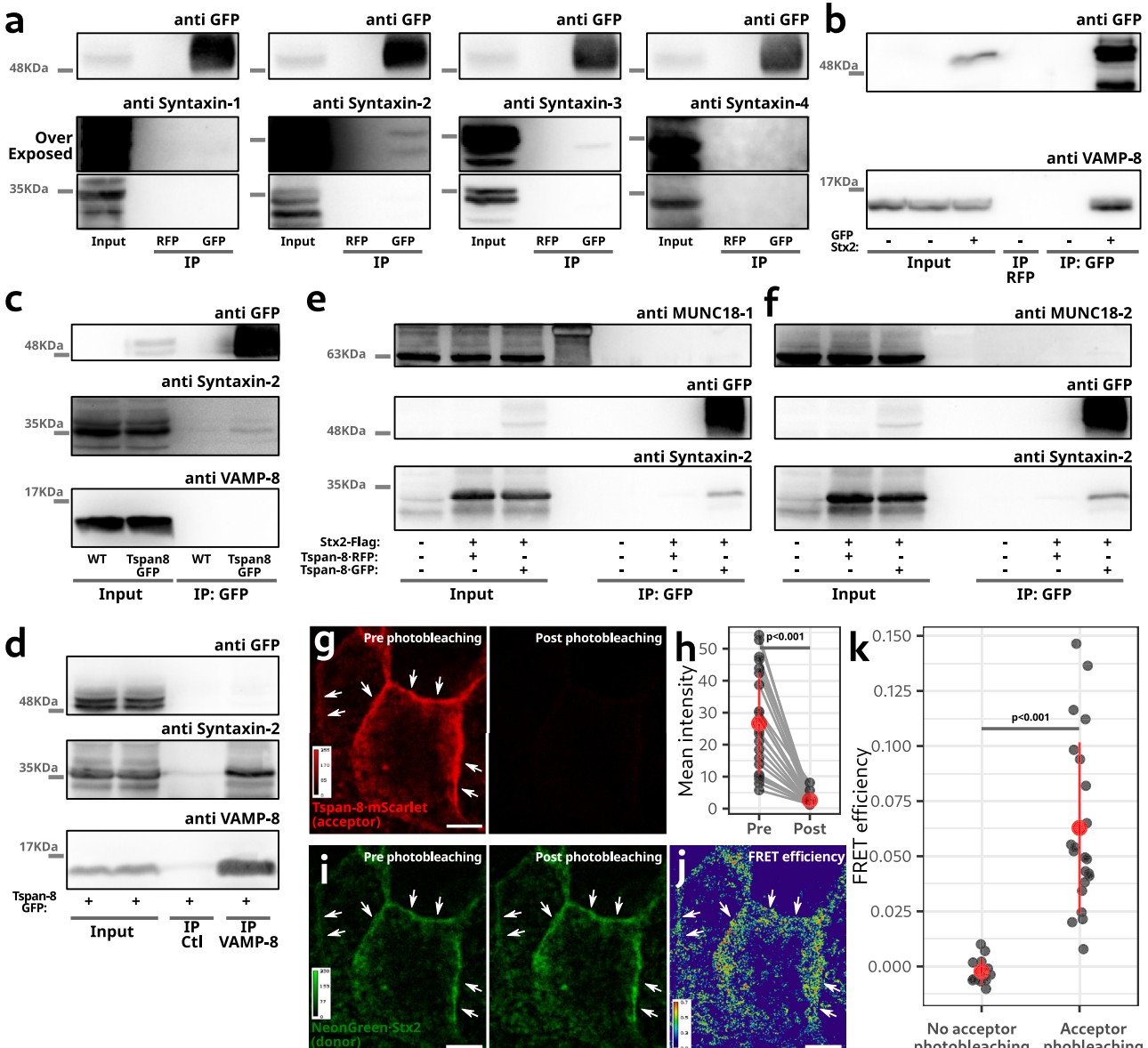

**Fig. 4 | Tspan-8 binds with syntaxins 2 and 3. a** Tspan-8·GFP immunoprecipitation and western blot analysis. Top panels show immunoblotting against GFP to confirm immunoprecipitation. Over-exposed membranes for better visualization of the co-precipitate are shown. RFP immunoprecipitation was used as control condition. **b** GFP-Stx2 immunoprecipitation and western blot analysis. Untransfected cells and RFP immunoprecipitation were used as control conditions. **c** Tspan-8·GFP immunoprecipitation and western blot analysis. WT HT29-N2 cells were used as a control condition. **d** Lysates of HT29-N2 cells genetically modified to express Tspan-8·GFP were processed for VAMP-8 immunoprecipitation and western blot analysis. Uncoated beads were used as control. **e, f** Tspan-8·GFP immunoprecipitation and western blot analysis. Top panels show immunoblotting against Munc18-1 (**e**) and Munc18-2 (**f**). No transfection and Tspan-8·RFP transfection were used as control conditions. **g** Representative images of the FRET acceptor before and after photobleaching. Arrows indicate regions of the plasma membrane. Scale bar is 5 μm. **h** Quantification of the fluorescence intensity of the acceptor pre and post

photobleaching. Same-images before and after photobleaching are linked with a straight line. Each dot represents the signal intensity of one image. *n* = 22. Images come from three independent cell cultures. Red dots represent the mean +/− the standard deviation. One-way ANOVA test: *p* < 0.001. **i** Representative images of the FRET donor before and after photobleaching of the FRET acceptor. Arrows indicate regions of the plasma membrane. Scale bar is 5 μm. **j** FRET map calculated with images shown in **i**. Arrows indicate regions of the plasma membrane with high FRET. FRET efficiency was color-coded with a pseudo-color look up table shown in the image. Scale bar is 5 μm. **k** Quantification of the FRET efficiency between NeonGreen·Stx2 and Tspan-8·mScarlet. Each dot represents the FRET efficiency of one image. Control conditions are images for which no bleaching step was performed. 13 control and 22 bleached images were analyzed. Red dots represent the mean +/− the standard deviation. One-way ANOVA test: *p* < 0.001. Source data for a to k are provided as a Source Data file.

in PCC towards the apical region is a reflection of the poor quality of the anti Stx2- antibody evidenced by its labelling of the nucleus in the basal region of cells (Fig. 5a, b, d, e).

*STX2* deletion (Fig. 5g (protein) and Supplementary Fig. 6a (mRNA)) did not affect total mucin-5AC levels in the cell lysates (Fig. 5h). A secretion assay showed that *STX2* removal reduced the ATP-stimulated secreted mucin-5AC·GFP by 64% (Fig. 5i, j). *STX2* RNAi

treatment in WT HT29-N2 cells confirmed this finding (Fig. 6f, g). Basal mucin secretion was also decreased in the *STX2* KO (Fig. 5i, j). Quantification of beta-tubulin in the cell lysates by western blot shows equal amount of cells in these secretion assays (Fig. 5g and Supplementary Fig. 6b and c).

These data confirm the involvement of Stx2 in basal and ATP-stimulated mucin secretion. The residual amount of mucin secretion in

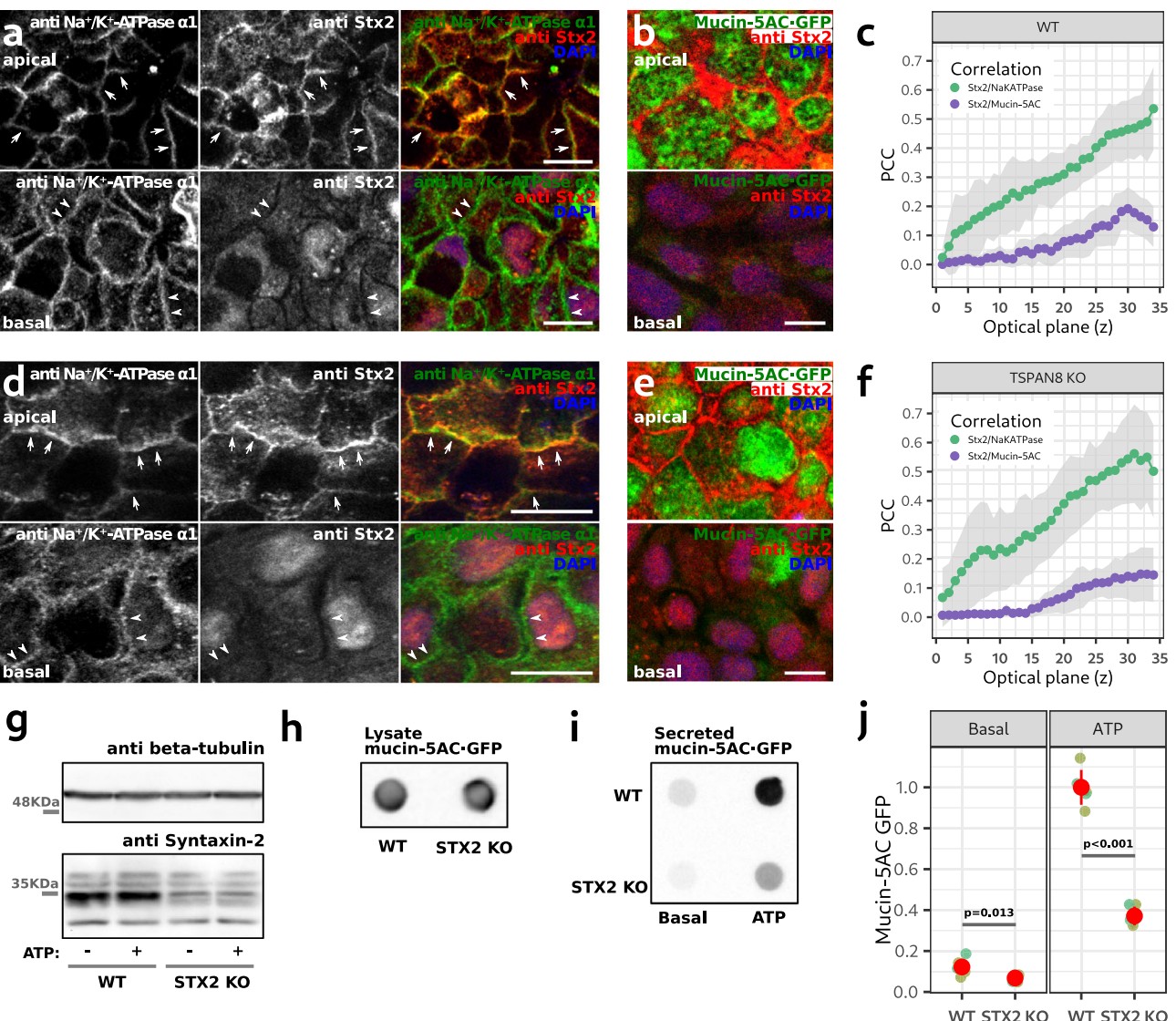

**Fig. 5 | Syntaxin-2 is necessary for mucin-5AC secretion. a** Optical planes of a representative confocal image of a HT29-N2 WT/Caco2 co-culture immunolabelled for the Na⁺/K⁺-ATPase α1 and Stx2. Arrows point to regions of the apical plasma membrane where Na⁺/K⁺-ATPase α1 and Stx2 co-localize. Arrowheads point to the basal plasma membrane. Scale bar is 10 μm. **b** Optical plane of a representative confocal image of a HT29 WT/Caco2 co-culture expressing mucin-5AC·GFP and immunolabelled for Stx2. Scale bar is 10 μm. **c** Quantification of the PCC between Na⁺/K⁺-ATPase α1 and Stx2 in WT cells. Green dots and lines show the mean PCC quantification between the Na⁺/K⁺-ATPase α1 and Stx2 signals in each optical plane. Purple dots and lines show the mean PCC between Stx2 and mucin-5AC·GFP. The gray areas show the mean PCC +/− the standard deviation. **d**. Same as in A for *TSPAN8* KO cells. **e** Same as in B for *TSPAN8* KO cells. **f** Same as in C for *TSPAN8* KO cells. **g** Representative western blot of the total amount of beta tubulin and Stx2 in WT and *STX2* KO cells corresponding to the mucin secretion assay shown in **i**. **h** Representative dot blot showing the total mucin-5AC·GFP content in differentiated WT and *STX2* KO cell lines. **i** Representative dot blot of a secretion assay of WT and *STX2* KO cells. **j** Quantification of secretion assays of WT and *STX2* KO cell lines. Each dot represents the mucin-5AC·GFP signal from one secretion assay. Grouped in different colors are secretion assays that were processed in parallel. *n* = 9. Red dots represent the mean +/− the standard deviation. Values are expressed as relative to the average mucin-5AC·GFP signal in ATP-stimulated WT cells. Statistical analysis was performed independently for basal and ATP-stimulated conditions. One-way ANOVA analysis: basal *p* = 0.013; ATP *p* < 0.001. Source data for **c** and **f** to **h** are provided as a Source Data file.

the *STX2* KO cells suggests some level of redundancy with another syntaxin. Brunger and colleagues have recently shown in an in vitro system that Stx3 can mediate vesicle fusion provided some accessory proteins are present[43]. Although syntaxins in vitro can exhibit some degree of promiscuity for its binding partners[44,45], the data of Brunger and colleagues suggest that Stx3 potentially function in the process of mucin granule fusion, but this remains to be formally tested in cells.

**Syntaxin-2 and Tspan-8 are in the same pathway of regulated mucin secretion**

Tspan-8 and Stx2 both localized to the plasma membrane (Figs. 3 and 5) and co-immuno-precipitated (Fig. 4). Co-transfection of

Stx2·RFP and Tspan-8·GFP and quantification of the PCC between them confirmed they co-localize at the plasma membrane (Fig. 6a, b). Confocal microscopy of the CRISPR/Cas c-terminally tagged Tspan-8·GFP, immunolabeling of endogenous Stx2 and the lipophilic plasma membrane-marker CellBrite® further confirms the co-localization of Stx2 and Tspan-8 at the plasma membrane of mucin-secreting cells (Fig. 6c). Co-lozalization between Stx2 and Tspan-8 at the plasma membrane was also seen by confocal microscopy of live cells stably over expressing Tspan-8·mScarlet and GFP·Stx2 by lentiviral infection (Supplementary Video 6).

But do these proteins function in the same linear pathway controlling the fusion of mucin-filled granules? *STX2* RNAi in WT and

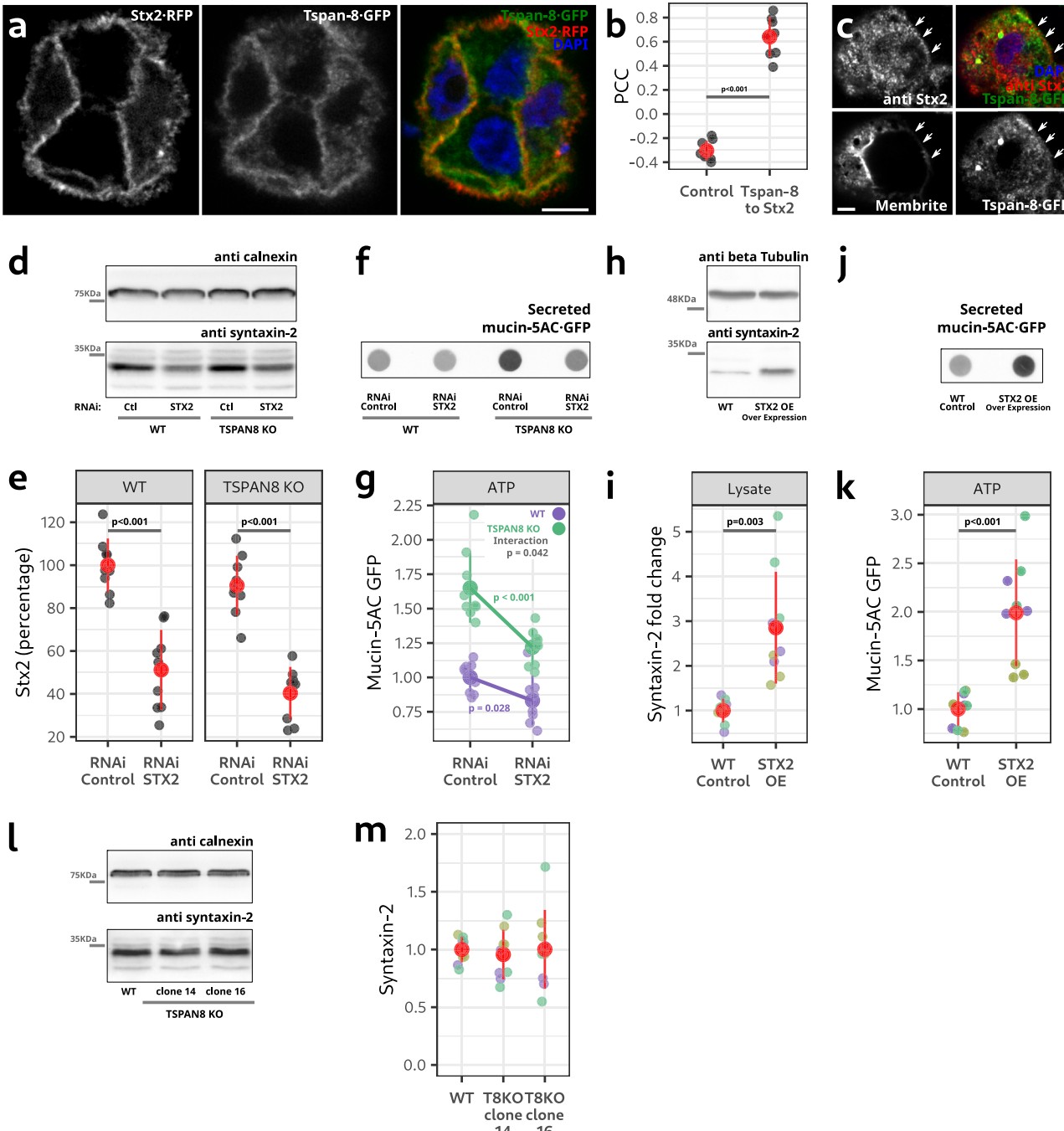

**Fig. 6 | Syntaxin-2 and Tspan-8 are in the same pathway of regulated mucin secretion. a** Cells co-transfected with Stx2·RFP and Tspan-8·GFP. Scale bar is 10 μm. **b** Quantification of the PCC between Stx2·RFP and Tspan-8·GFP. Each gray dot represents the PCC of one image. *n* = 8. One-way ANOVA *p* < 0.001. **c** Cells immunolabeled with an anti Stx2 antibody, Tspan-8·GFP and CellBrite®. Arrows point to the plasma membrane. Scale bar is 5 μm. One image of three independent cultures is shown with similar results in all images. **d** Western blot of Stx2 in WT and *TSPAN8* KO cells treated with RNAi control or against *STX2*. **e** Quantification of the amount of Stx2 in WT and *TSPAN8* KO cells treated with RNAi control or against *STX2*. Each dot represents Stx2 signal from an independent sample. *n* = 9. Two-way ANOVA test: genotype *p* = 0.146; RNAi treatment *p* value < 0.001. TukeyHSD *p* values shown. **f** Dot blot showing mucin-5AC·GFP secretion from WT and *TSPAN8* KO cells treated with RNAi control or against *STX2*. **g** Quantification of mucin-5AC·GFP in WT

and *TSPAN8* KO cells treated with RNAi control or against *STX2*. Each dot represents the signal from 1 secretion assay. *n* = 9. Two-way ANOVA test: genotype: *p* < 0.001; RNAi treatment *p* < 0.001; interaction *p* = 0.042. *p* values of a one-way ANOVA for each genotype are shown. **h.** Western blot of Stx2 in WT and *STX2*-over expressing cells. **i** Quantification of Stx2 in WT and *STX2*-over expressing cells. Each dot represents Stx2 from an independent sample. *n* = 9. One-way ANOVA *p* = 0.003. **j** Dot blot of a secretion assay showing secreted mucin-5AC·GFP of ATP-stimulated WT and *STX2*-over expressing cells. **k** Quantification of the secretion assays of WT and *STX2*-over expressing cells. Each dot represents the mucin-5AC·GFP from one assay. *n* = 9. One-way ANOVA *p* < 0.001. **l** Western blot of Stx2 in WT and *TSPAN8* KO cells. **m** Quantification of the amount of Stx2 in WT and *TSPAN8* KO cells. Each dot represents the Stx2 signal from an independent sample. *n* = 9. Source data for b and d to m are provided as a Source Data file.

*TSPAN8* KO cells decreased *STX2* expression between 50 and 60% on average (Fig. 6d, e). Acute down regulation of *STX2* by RNAi did not affect mucin-5AC·GFP expression (Supplementary Fig. 7a–d). We found that *STX2* downregulation differentially decreased mucin-5AC·GFP release in WT and *TSPAN8* KO cells (Fig. 6f, g—ANOVA interaction *p* value = 0.042; Loading controls in Supplementary Fig. 7e and f) suggesting that both proteins function in the same pathway leading to mucin secretion.

To provide further functional evidence that the amount of Stx2 available at the plasma membrane regulates the quantity of mucin release, we over expressed *STX2* with a lentiviral system in mucin-secreting cells. Puromycin-selected and differentiated cells had an average 3-fold increase in Stx2 amount (Fig. 6h, i). *STX2* over expression did not change mucin-5AC·GFP expression (Supplementary Fig. 7g–j) but doubled the amount of mucin-5AC·GFP secreted in ATP-stimulated cells (Fig. 6j, k; Loading controls in Supplementary Fig. 7k and l). Since over expression of *STX2* increases mucin release we determined by western blot whether *TSPAN8* KO affected *STX2* expression. We found that Tspan-8 ablation did not change Stx2 levels (Fig. 6l, m).

## Mutant Tspan-8 is retained in the ER, sequesters Stx2 and inhibits mucin5-AC secretion

By CRISPR/Cas genome editing we modified the *TSPAN8* locus in mucin-secreting cells to expresses a c-terminal truncated version of Tspan-8 (d234-237). Imaging of cells revealed that mutant Tspan-8 d234-237 is retained in the endoplasmic reticulum (ER) and its availability at the plasma membrane is therefore reduced (Fig. 7a, b). Fluorescence microscopy revealed that under these conditions, Stx2 is also localized to the ER and the amount of Stx2 at the plasma membrane is concomitantly reduced (Fig. 7c). We immunprecipitated the mutant Tspan-8 and immunoblotted the precipitate with an anti Stx2 antibody. Our data reveal that mutant Tspan-8 binds Stx2 (Fig. 7d). A mucin secretion assay showed that the cell line expressing the mutant Tspan-8 d234-237 secretes on average 50% less mucin5-AC compared to WT cells (Fig. 7e, f).

These results show that: (1) The c-terminus of Tspan-8 is necessary for its export from the ER; (2) The c-terminus of Tspan-8 is dispensable for Stx2 interaction; (3) The relocation of Stx2 in Tspan-8 d234-237 KI cells supports the interaction between the two proteins shown by biochemical methods (Fig. 4a, c, e, f; coIPs) and microscopy

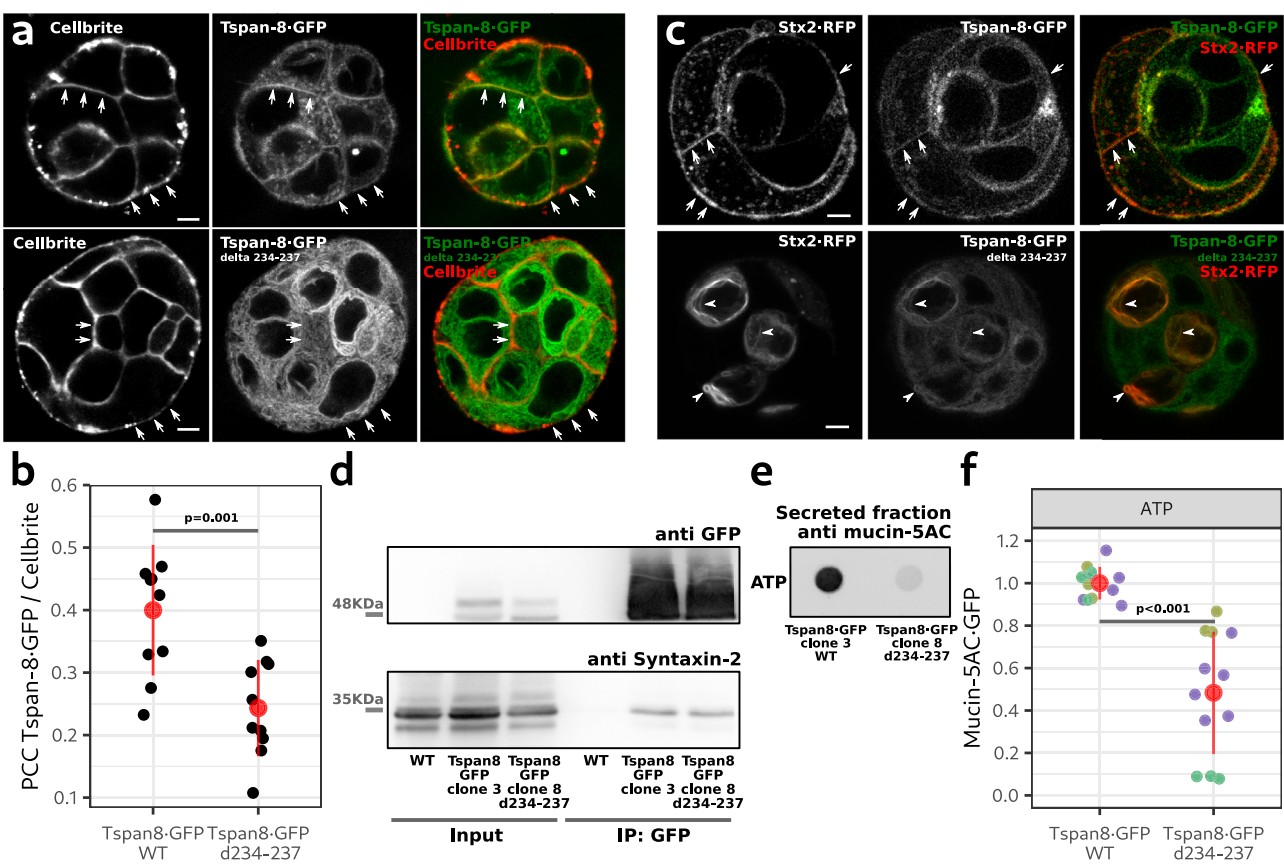

**Fig. 7 | Tspan-8 d234-237 is arrested in the ER, sequesters Stx2 and inhibits mucin5-AC·GFP secretion. a** Optical plane from a confocal image of live cells expressing Tspan-8·GFP and labelled with CellBrite® to visualize the plasma membrane. Top row are cells expressing WT Tspan-8 (clone 3) and lower row are cells expressing the truncated version of Tspan-8 d234-237 (clone 8). Arrows point to the plasma membrane. Scale bar is 5 μm. **b** Quantification of the PCC between Cellbrite® and Tspan-8·GFP. Each gray dot represents the PCC of 1 image with at least 3 cells. *n* = 10. The red dot is the mean value of the gray dots +/− the standard deviation. The *p* value was calculated with an ANOVA test. **c** Optical plane from live cells expressing Tspan-8·GFP at endogenous levels and transiently transfected with Stx2·RFP. Top row are cells expressing WT Tspan-8·GFP (clone 3) and lower row are cells expressing the truncated version of Tspan-8 d234-237 (clone 8). Images of clone 3 (*n* = 8) and clone 8 (*n* = 11) were taken from two independent cell cultures.

All images showed similar results. Arrows point to the plasma membrane. Arrowheads point to Stx2·RFP trapped in the ER. Scale bar is 5 μm. **d** Lysates of HT29-N2 cells genetically modified by CRISPR/Cas9 to express WT (clone 3) or a truncated version (clone 8) of Tspan-8·GFP were processed for immunoprecipitation of GFP and western blot analysis. WT HT29-N2 cells were used as a control condition. One of two independent experiment is shown. Both experiments showed similar results. **e** Representative dot blot showing mucin-5AC secreted by cells expressing WT or a truncated version of Tspan-8. **f** Quantification of the amount of secreted mucin-5AC in cells expressing the WT or the truncated version of Tspan-8. Each dot represents the signal from one secretion assay. Grouped in different colors are secretion assays that were processed in parallel. *n* = 12. Red dots represent the mean +/− the standard deviation. One-way ANOVA test: *p* < 0.001. Source data for **b**–**f** are provided as a Source Data file.

(Fig. 4g, j, k; FRET); (4) The defect in mucin secretion in Tspan-8 d234-237 is likely due to reduction in the levels of Stx2 at the plasma membrane. Altogether, the data shown in Figs. 6 and 7 strongly suggest that Tspan-8 and Stx2 work in the same pathway and that their interaction is necessary for their role in mucin secretion.

## Tspan-8 over expression inhibits glucose-stimulated insulin secretion from INS-1 cells

Our data shows that loss of *TSPAN8* increases mucin secretion. It is therefore reasonable to assume that over expression of *TSPAN8* would inhibit mucin secretion or secretion of another molecule that might utilize the same principle for its release by cells. We therefore chose to test the effect of *TSPAN8* expression on insulin secretion. Insulin secretion is also mediated by large granules that undergo condensation, is shown to be dependent on syntaxin-1[46], and cell lines are amenable to experimental manipulation. Mice insulin-secreting cells do not express *TSPAN8*[47], but it is likely that a related tetraspanin plays a similar role to tetraspanin-8 in mucin secretion. For example, the colon cells express TRPM5 for mucin secretion, whereas airway cells express TRPM4 for the same function[3].

By western blot analysis we confirmed that rat insulin-secreting cells (INS-1) do not express *TSPAN8* (Fig. 8a). We over expressed WT human Tspan-8 c-terminally tagged with mScarlet in rat INS-1 cells by lentiviral infection and selected positive cells by FACS to generate a stable cell line (Fig. 8b). Confocal microscopy of live INS-1 cells also confirmed Tspan-8·mScarlet over expression and showed that it localized to the plasma membrane and in some intracellular structures (Fig. 8c; arrows and arrowheads respectively) as in HT29-N2 cells (Fig. 3c). Immunoprecipitation of Tspan8·mScarlet revealed the presence of syntaxin-1 in the precipitate (Fig. 8d). An insulin secretion assay in the presence of glucose in the culture media was used to measure insulin secretion by ELISA. This experiment showed that Tspan-8 over expression in INS-1 cells reduced the amount of glucose-stimulated insulin secretion (Fig. 8e). These data support the general role of tetraspanins and syntaxins at the plasma membrane in controlling release propensity of granules and raises the question of how Tspan-8 binds different syntaxins depending on the cell type.

## Discussion

There is a general lack of reagents and systems amenable to standard molecular biology approaches to unravel the mechanism of mucin secretion. This is why we know so little about how mucins are sorted and exported from the ER and the Golgi, how they are packed into micro meter size granules, how granules condense and only a subset fuses either by the baseline or the stimulated pathway. Our reductionist approach of using a cancer-derived, mucin secreting cell line comes at a cost that it might not reveal the entire picture as in a more physiologically relevant system. But the advantage of this reductionist system is that it leads to the identification of proteins which can then be tested for their physiological significance.

An understanding of the mechanisms by which cells control the release propensity, particularly in the two phases of stimulated secretion, is an important unaddressed issue. Our data show that *TSPAN8* expression is upregulated in mucin-secreting cells and that Tspan-8 functions at the plasma membrane to control the release propensity of mucin granules during the slow and sustained phase of ATP-stimulated secretion. Identification of tetraspanin-8 in controlling mucin release by sequestering syntaxin-2 answers the long-standing question of how cells regulate the external agonist-stimulated, biphasic release of secretory cargoes.

## Tspan-8 controls the number of mucin granules that can dock for fusion

External agonist-stimulated fusion of secretory granules to the plasma membrane is biphasic. The first high-rate secretion phase lasts a few seconds and the second phase has a lower release rate but can last several minutes[17,46]. Docking is a fundamental process preceding vesicle fusion that is mediated by syntaxins[29–31]. It has been suggested that docking of synaptic vesicles during sustained neurotransmitter release is the rate limiting step[20].

Our data reveal that stimulated mucin release is biphasic and that tetraspanin-8 is crucial to control the release propensity of granules in reserve. *TSPAN8* KO increased mucin release during the second phase of secretion (Fig. 2). Syntaxin-2 localization and expression are not affected by *TSPAN8* KO (Figs. 5 and 6), therefore

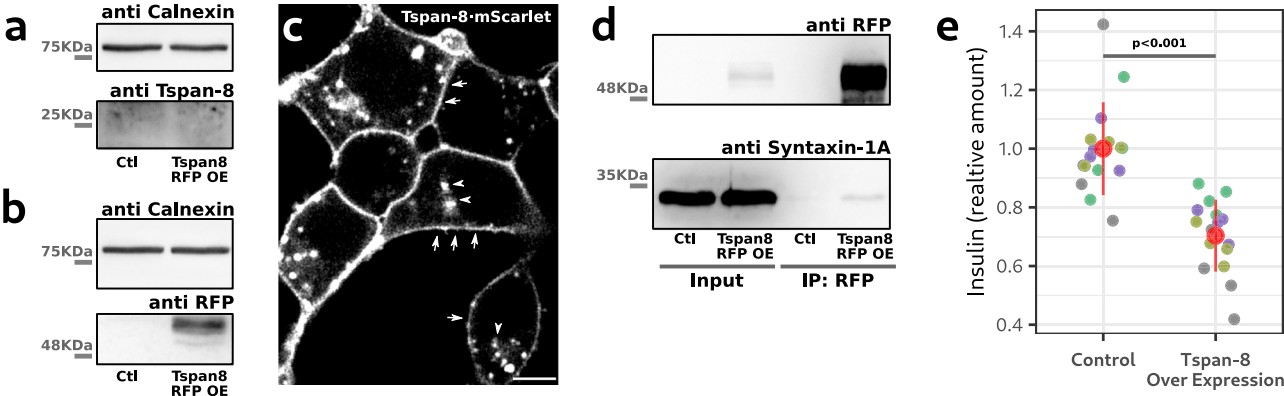

**Fig. 8 | Tspan-8 over expression inhibits insulin secretion. a** Western blot of the total amount of Tspan-8 (~25 KDa) in INS-1 cells. OE means over expression and Ctl refers to WT INS-1 cells. One image of four independent experiments is shown. Results are similar in all replicates. Not shown replicates are provided in the source data file. **b** Western blot of the total amount of human Tspan-8·mScarlet (~51 KDa) in INS-1 cells. Membranes were immunoblotted with anti calnexin and anti RFP antibodies, respectively and developed by ECL. OE means over expression and Ctl refers to WT INS-1 cells. One image of four independent experiments is shown. Results are similar in all replicates. Not shown replicates are provided in the source data file. **c** Optical plane from a confocal image of live INS-1 cells over expressing human Tspan-8·mScarlet. Arrows point to the plasma membrane. Arrowheads point to intracellular human Tspan-8·mScarlet. Scale bar is 5 µm. One of six

independent images is shown. Images were taken from two independent cell cultures. All images showed similar results. **d** Lysates of WT or hTspan-8·mScarlet-over expressing INS-1 cells were processed for immunoprecipitation of RFP and western blot analysis. OE means over expression and Ctl refers to WT INS-1 cells. One image of two independent experiments is shown. Results are similar in all replicates. **e** Quantification of glucose-stimulated insulin secretion in WT and hTspan-8·mScarlet-over expressing INS-1 cells. Insulin in the culture media was measured by ELISA. Values are expressed as relative to the control condition. Grouped in different colors are secretion assays that were processed in parallel. Red dots represent the mean +/− the standard deviation. Total number of replicates are 15. The *p* value (<0.001) of a one-way ANOVA analysis is shown in the plot. Source data for **a**, **b**, **d** and **e** are provided as a Source Data file.

Tspan-8 controls the quantities of syntaxin-2 available at the plasma membrane for docking of granules (Fig. 9). Tspan-8 localizes to the plasma membrane of secretory cells (Fig. 3), co-localizes with Stx2 (Fig. 6), and co-immunoprecipitates with syntaxins 2 and 3 in HT29-N2 cells (Fig. 4). Furthermore, we show that Stx2 can form a complex with either VAMP-8 or Tspan-8, but not both (Fig. 4). The Tspan-8/Stx2 complex does not contain VAMP-8 or Munc18 (Fig. 4). This suggests that ordinarily, Tspan-8 regulates secretion by binding or sequestering Stx2 (and perhaps Stx3) to effectively limit the available Q-SNAREs at the plasma membrane and keeping docking sites empty and refractory. Cells lacking Tspan-8 provide quantitatively more Stx2 for docking and therefore increases the fusion of mucin granules. The kinetics of the fast-rate phase of mucin release in WT and *TSPAN8* KO cells are similar (Fig. 2). Tspan-8 therefore does not affect mucin granules that are docked before ATP stimulation but prevents the docking of a pool of granules that are kept in reserve (Fig. 9). This also explains why Tspan-8 loss does not affect baseline mucin secretion.

Like Mucin, insulin secretion is also biphasic[46]. Tspan-8 is expressed in the human pancreas[47] and allelic variants of Tspan-8 have been associated with type-II diabetes mellitus[48,49]. Mouse beta cells on the other hand, do not express Tspan-8 and a KO model showed no defect in insulin secretion[47]. It is conceivable that a different tetraspanin performs the function of Tspan-8 in mice. This is supported by our data for insulin secretion by rat insulin secreting (INS-1) cells. Tspan-8 over expression in INS-1 cells affected glucose-dependent insulin secretion (Fig. 8). Tspan-8 could be precipitated with Stx1 (Fig. 8). How the differential specificity in binding in different cell types is achieved is an important challenge and will emerge from identifying the binding sites and the involvement of other proteins that aid in these affinities.

### A general mechanism for docking granules in reserved pool

Our reductionist cell line-based approach has revealed tetraspanin-8 as a new component in the overall scheme that controls quantities of mucins secreted. The challenge now is to test how tetraspanin-8 functions in the physiological setting of the colon and airways where other cells such as sentinel and ciliated (missing in our methods used here) function with goblet cells to maintain extracellular mucin levels. Our findings raise the challenge on the exact meaning of Tspan-8 and Stx2 interaction. A simple way to look at this is that Tspan-8 organizes docking sites by sequestering proteins like Stx2 in microdomains of the apical plasma membrane, thereby limiting their supply during the second-slower phase of the agonist-mediated release of granule content (Fig. 9). Why *TSPAN8* sequesters only a subpopulation of plasma membrane specific syntaxins and functions specifically in the sustained release step of agonist dependent cargo release are important issues to address? Our data that Tspan-8 binds Stx1 in INS-1 cells to handle insulin secretion and Stx2 in HT29-N2 to affect mucin exocytosis suggests a general role for Tspan-8, and perhaps other tetraspanins, in regulating biphasic cargo release in a cell specific manner.

## Methods

### Cell culture and differentiation

HT29-N2 cells (obtained from ATCC) (RRID: CVCL_5942), were grown in Dulbecco's Modified Eagle's Medium (DMEM) cell culture media (Lonza. Cat. No.: BE12-604F/U1) supplemented with 10% (vol / vol) heat-inactivated fetal bovine serum (Gibco. Cat. No.: 10270-106) (complete media). Cells were passaged at a 1:4 ratio every Monday and Friday. Cells were kept at 37 °C in a humidified incubator supplied with $CO_2$ 5%. Validation of the cell line was done by determining the production and secretion of mucin5-AC by dot blot analysis.

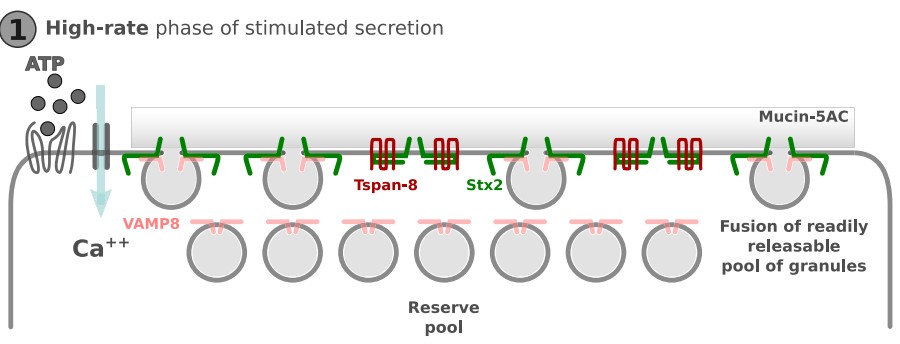

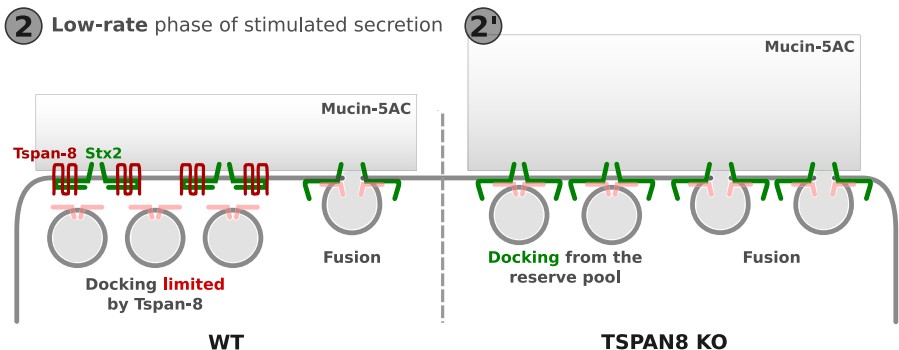

**Fig. 9 | Model for function of Tspan-8 to regulate biphasic release of granules.** **1.** Pre-docked granules fuse upon exposure of cells to external agonist like ATP. **2.** Tspan-8 at the plasma membrane binds Stx2 preventing it from docking granules from the reserve pool during the second phase of stimulated secretion. **2′.** Loss of Tspan-8 exposes Stx2 to engage with granules from the reserve pool that fuse to increase the quantity of mucins secreted.

For differentiation, $2 \times 10^5$ cells/cm$^2$ were plated into a T25 or T75 culture flask with complete media, typically on a Monday. The following day, cells were washed with phosphate buffered saline (PBS) and complete media was changed with Protein-Free Hybridoma Media (PFHM) (Gibco. Cat. No.: 12040-051). After 3 days, media was replaced with fresh PFHM. After 6 days in culture, differentiated cells were trypsinized (Gibco. Cat. No.: 25300-054). For secretion assays and live-imaging experiments, $1.1 \times 10^5$ cells/cm$^2$ were plated in complete media in 6- or 12-well plates. For live imaging, cells were seeded in glass-bottom 35 mm dishes (ibidi. Cat. No.: 81156). For immuno-fluorescence, cells were plated at $1.7 \times 10^4$ cells/cm$^2$ in culture dishes containing cover-slips for microscopy. For immunoprecipitation experiments $2 \times 10^5$ cells/cm$^2$ were plated in 60 mm dishes. The following day, cells were washed with PBS and complete media was changed with PFHM[33,34].

## HT29-N2 / Caco-2 co-culture

Caco2 (ATCC - HTB-37) and HT29-N2 co-culture were prepared as previously described[42], with small variations. Caco2 cells were cultured in DMEM 4.5 g/L Glucose with UltraGlutamine (Lonza: H3BE12-604F/U1) medium containing 20% fetal calf serum (Gibco: 10270106), supplemented with 100 units/mL penicillin, and 100 µg/mL streptomycin (Invitrogen: 15070063). Cells were incubated at 37 °C in a humidified incubator supplied with 5% $CO_2$. Caco-2 and HT29-N2 cells were mixed prior to seeding in a ratio of 50:1 (Caco-2/HT29-N2) at a density of 2.5 $10^5$/cm$^2$. The co-culture was maintained in DMEM 4.5 g/L Glucose with UltraGlutamine medium supplemented with 20% fetal calf serum, and the culture medium was changed every 2–3 days. Images of polarized HT29-N2 clusters were taken between 14 and 21 days after seeding.

## INS-1 culture

Rat insulinoma cell line (INS-1. Sigma-aldrich - SCC208) was grown in Roswell Park Memorial Institute (RPMI) cell culture media (Thermo-Fisher Scientific. Cat. No.: 1640) supplemented with 10% (vol/vol) heat-inactivated fetal bovine serum (Gibco. Cat. No.: 10270-106), 10 mM HEPES, penicillin 50 units/ml – streptomycin 50 µg/ml (ThermoFisher Scientific. Cat. No.: 15070063), 50 µM 2-mercaptoetanol (ThermoFisher Scientific. Cat. No.: 31350010) and 1 mM sodium pyruvate (Sigma-Aldrich. Cat. No.: S8636). Cells were passaged at a 1:3 ratio every Monday and Friday. Cells were kept at 37 °C in a humidified incubator supplied with $CO_2$ 5%. Validation of the cell line was done by determining the production and secretion of insulin by ELISA. For insulin secretion assays, cells were trypsinized (Gibco. Cat. No.: 25300-054), determined their density with a Neubauer chamber and $0.25 \times 10^6$ cells were seeded in a 24-well plate. Cells were kept at 37 °C in a humidified incubator supplied with $CO_2$ 5% for 2–4 days before the assay.

## Cell transfection

Cells were transfected for transient plasmid over expression with lipofectamine (ThermoFisher Scientific. Cat. No.: L3000001) as previously described[50] at the moment of plating. For immuno-fluorescence experiments, cells were transfected in 35 mm in diameter cultures dishes with 2 ml of complete culture media. For each transfection reaction 7 µl of lipofectamine were diluted in 125 µl of OptiMEM and between 1.5 and 3 µg of plasmid DNA were diluted in 125 µl of OptiMEM. The lipofectamine and DNA solution were then mixed incubated at RT for 15 min and added to the culture dish. For immuno-precipitation transfections the ratio of lipofectamine to DNA were maintained but scaled up to 250 µl for each reaction transfection.

## RNA interference transfection

Cells were transfected for RNA interference with lipofectamine RNAi-MAX (ThermoFisher Scientific. Cat. No.: 13778075). HT29-N2 cells were differentiated and seeded in 24-well plates. At the moment of seeding, the first RNAi transfection mix was added to the culture media.

For each reaction, the transfection mix was prepared by diluting 3.125 µl of lipofectamine in 50 µl of OptiMEM. The RNAi mix was prepared by diluting two independent RNAi against *STX2* down to 20 nM each one. Sequences are shown in Supplementary Table 1. The lipofectamine and the RNAi solutions were mixed and incubated at RT for 10 min. The mix was then added to the culture well. Two days after the first reaction, a second, identical transfection was performed and 4 days after the first transfection, the secretion assay were performed.

## CRISPR/Cas9 genetic engineering

To generate Knock-Out (KO) and Knock-In (KI) cell lines we used the RNA-guided Cas9 endonuclease system developed from the microbial clustered regularly interspaced short palindromic repeats (CRISPR) adaptive immune system[51]. Guide RNA sequences were chosen with the web-based tool CRISPOR which implements scoring algorithms based on their potential off target and on-target DNA cleavage activity[52]. Sequences are shown in Supplementary Table 2. The highest-scored guide RNA sequences were cloned into a pSpCas9(BB)-2A-GFP backbone plasmid. pSpCas9(BB)-2A-GFP (PX458) was a gift from Feng Zhang (Addgene plasmid #48138; http://addgene.org/48138; RRID: Addgene_48138)[51]. After confirmation of correct insertion of the guide RNA oligonucleotides by colony PCR and sequencing, plasmids were amplified, purified with a midi prep kit and transfected to HT29-N2 cells with lipofectamine 3000 as described in the cell transfection section of materials and methods. Two to 3 days after transfection, cells were trypsinized and detached from the culture dish, pelleted and re-suspended in complete culture media. Cells were then sorted by a Fluorescence Activated Cell Sorter (FACS). For KO generation 1 GFP-positive cell was placed in each wheel of a 96 multi-well plate containing complete media supplemented with plasmocin (15 µg/ml) (InvivoGen. Cat. No.: ant-mpt). Cells were amplified until final confirmation of gene knockout by RT-PCR and protein electrophoresis followed by western blot as previously described[53]. To increase the chances of successful gene KO 3 different guide RNAs targeting the ATG origin of translation, middle and c-terminus regions of the target locus were co-transfected. For gene KI, 1 guide RNA (designed to target the c-terminus of the locus) was co-transfected with the PCR product of the sfGFP sequence amplified with primers bearing tails complementary to the upstream and downstream regions of the expected cut site by Cas9 in the target gene. Two to 3 days after transfection, GFP-positive cells were bulk-collected by FACS and grown for 10–15 days in complete media. After this time, GFP signal from the CRISPR/Cas9 plasmid has diluted away and any GFP signal should come from the integration of the transfected DNA template into the target genome. Cells were then sorted again by FACS and 1 GFP-positive cell was placed in each wheel of a 96 multi-well plate containing complete media supplemented with plasmocin (15 µg/ml). Cells were amplified until final confirmation of gene KI by PCR and protein electrophoresis followed by western blot.

**Generation of stable over expressing cell lines.** Lentiviral particles were generated by co-trasnfecting pRSV–REV, pMDLg/pRRE, VSV-G [54] and the transfer plasmid (L309) carrying the sequence of the gene to be over expressed with resistance to puromycin into $0.8 \times 10^6$ HEK−293T (ATCC, negative for mycoplasma) cells in a 60 mm culture dish with 6 µl of TrasnIT (Mirus Bio. Cat. No.: 293) containing 3 ml of complete media. The following day the cell culture media was changed with fresh media. Two and three days after, the cell media was collected and filtered through a 0.45 µm membrane. HT29-N2 cells were plated in 60 mm dishes. The following day the media was changed for 2 ml of fresh complete media, 1 ml of the filtered viral particles and 8 µg/ml of hexadimethrine bromide (polybrene) (Sigma-Aldrich. Cat. No.: 107689). The following day media was change with fresh media. Two days after, the culture media was changed for complete media supplemented with puromycin (Gibco. Cat. No.: A11138-03) 15 µg/ml for

1 week. After this time, cells were maintained in compete media supplemented with puromycin 2.5 µg/ml.

If the gene to be over expressed had a fluorescent tag (GFP·Stx2 or Tspan-8·mScarlet), the transfer plasmid did not contain the puromycin resistance gene and selection was achieved by Fluorescent Activated Cell Sorting.

## Mucin secretion assays

Secretion assays were done as previously described [3,34,55,56]. Culture media of differentiated HT29-N2 cells was replaced with 0.5 ml (12-well plates) or 1 ml (6-well plates) of isotonic buffer (KCl 2.5 mM; NaCl 140 mM; CaCl$_2$ 1.2 mM; MgCl$_2$ 0.5 mM; Glucose 5 mM; HEPES 10 mM; pH 7.42 adjusted with tris base and 305 mOsm/litre adjusted with D-manitol if needed). For stimulated secretion, the isotonic solution was supplemented with 100 µM ATP. Cells were kept with the isotonic solution at 37 °C for 30 min. The supernatant was collected and centrifuged for 5 min at $800 \times g$ at 4 °C to eliminate cells that might have detached during the procedure. 80% of the supernatant was recovered in a new tube (secreted fraction). Cells were lysed in lysis buffer as described in the western blot section of material and methods.

## Insulin secretion assays

Culture media of INS-1 cells was replaced with 0.25 ml of isotonic buffer and cells were kept at 37 °C for 45 min. The isotonic buffer was replaced with isotonic buffer supplmented with 2 mM D-(+)-Glucose (Sigma-Aldrich. Cat. No.: 49139) and cells were maintained at 37 °C for 45 min. The isotonic media was collected in a 1.5 ml tube and centrifuged 5 min at $800 \times g$ at 4 °C to eliminate cells that might have been detached during the procedure. The supernatant was collected and kept in a new 1.5 ml tube. Samples were frozen until determination of the amount of insulin by ELISA (Crystal Chem. Cat. No.: 90060). The determination of the amount of insulin in the samples by ELISA was done following the manufacturer's indications.

## Dot blot

100 microlitres of the secreted fractions and 100 µl of 1:20 diluted (in PBS) lysates from the secretion assays were loaded into a bio-blot microfiltration blotting system (BioRad. Cat. No.: 1706545). Samples were left to flow through a 0.45 µm nitrocellulose blotting membrane (Amersham Protran. Cat. No.: 10600002) by gravity [3,55]. Membranes were then blocked with a Tris-Buffered Saline (TBS) solution supplemented with 0.1% vol/vol polysorbate 20 (Sigma-Aldrich. Cat. No.: P1379) (TBST) and 2.5% weight/vol non-fat dry milk for 20–40 min at room temperature (RT) on a laboratory rocker. Primary antibodies were diluted in TBST/BSA 5% (weight/vol) and membranes were incubated with this solution overnight at 4 °C on a laboratory rocker. Fluorescent secondary antibodies (donkey anti rabbit IgG – Alexa Fluor 680 – Life technologies. Cat. No.: A10043 or donkey anti mouse IgG – Alexa Fluor Plus 800 – Invitrogen. Cat. No.: A32789) were diluted to 2 µg/µl in TBST/non-fat dry milk 2.5% and membranes were incubated with this solution for 60 min at RT on a laboratory rocker. Fluorescent signal was detected using the iBright imaging system (ThermoFisher Scientific) equipped with a high resolution (9.1 mega pixels) CCD camera and multiplexed laser excitation/emission filters.

## Immunoprecipitation

Cells were washed once with cold PBS and 1.2 ml of lysis buffer (Tris 50 mM; NaCl 150 mM; pH 7.3; Triton X-100 1% (Sigma-Aldrich. Cat. No.: T9284); Leupeptin 5 µM (Focus Biomolecules. Cat. No.: 10-1346); Aprotinin 2 µg/µl (Abcam. Cat. No.: ab146286); Pepstatin A 2 µg/µl (Panreac Química. Cat. No.: A2205)) was added. Cells were kept in the lysis buffer on ice for 10 min and on a laboratory rocker. Cells were then flushed by gentle pipetting, collected in 1.5 ml tubes and placed in a rotating wheel at 4 °C for 30 min. Samples were centrifuged at 12000 g at 4 °C for 15 min. 0.1 ml of the supernatant was recovered in a

new tube and 20 µl of 6X loading buffer (Tris – HCl 375 mM; SDS 12 %; Glycerol 60%; DTT 600 mM; Bromophenol Blue 0.6%) was added (input). One ml of the supernatant was recovered in a new tube containing 25 µl of GFP-Trap® Agarose (ProteinTech/ChromoTek. Cat. No.: gta)beads covalently attached to anti GFP nanobodies/VHH. Where applicable, RFP-Trap® Agarose (ProteinTech/ChromoTek. Cat. No.: rta) was used as a control of non-specific binging of proteins to the beads and/or antibodies. Samples were incubated in a rotating wheel for 60 min at 4 °C. Beads were precipitated by centrifugation at $2500 \times g$ for 5 min and washed with 500 µl lysis buffer 4 times. Beads were resuspendended in 30 µl 1.2X loading buffer (IP fraction).

## Protein electrophoresis and western blot

Cells were washed once with cold PBS and 0.5 ml (12-well plates) or 1 ml (6-well plates) of lysis buffer (Tris 50 mM; NaCl 150 mM; pH 7.3; Triton X-100 1% (Sigma-Aldrich. Cat. No.: T9284); Leupeptin 5 µM (Focus Biomolecules. Cat. No.: 10-1346); Aprotinin 2 µg/µl (Abcam. Cat. No.: ab146286); Pepstatin A 2 µg/µl (Panreac Química. Cat. No.: A2205); DTT 1 µM (Sigma-Aldrich. Cat. No.: 43815)) was added. Cells were kept in the lysis buffer on ice for 5 min and on a laboratory rocker. Cells were then flushed by vigorous pipetting (12-well plates) or scraped (6-well plates or 60 mm dishes), collected in 1.5 ml tubes and placed in a rotating wheel at 4 °C for 20–30 min. Samples were centrifuged at $12,000 \times g$ at 4 °C for 15 min. The supernatant was recovered in a new tube and 1:6 in volume of 6X loading buffer was added (lysate). Lysates with loading buffer were boiled at 95 °C for 5 min. Twenty to 40 µl of the lysate were loaded into a 12% polyacrylamide gel for SDS-PAGE electrophoretic (BioRad. Mini-PROTEAN) separation of proteins. Samples were run 20 min at constant 90 V then 110 V for 60–80 min. Proteins were transferred to a 0.45 µm PVDF membrane (Amersham. Cat. No.: 10600023) for 90 min at constant 400 mA in an ice-bucket. Membranes were blocked with a solution of 2.5% (weight/weight) of non-fat dry milk in TBST for 20 to 30 min at RT on a laboratory rocker. Primary antibodies were diluted in TBST supplemented with 5% BSA and membranes were incubated with this solution overnight at 4 °C. Membranes were washed 3 times, 5 min each. Horseradish root peroxidase-conjugated (Jackson ImmunoResearch) or fluorescently-conjugated (Invitrogen) secondary antibodies were diluted in TBST to 0.08 µg/µl and 0.2 µg/µl respectively. Membranes were incubated with the secondary antibody solution for 60 min at RT in a laboratory rocker. Membranes were washed 3 times, 5 min each with TBST. Enhanced chemiluminescence (ECL) signal was generated by incubating the membranes with Immobilon Forte Western HRP Substrate (Millipore. Cat. No.: WBLUF0100). ECL and fluorescent emission were digitalized with the iBright imaging system (ThermoFisher Scientific) equipped with a high resolution (9.1 mega pixels) CCD camera.

## Immunofluorescence

Cells were fixed for 15 min with a solution of paraformaldehyde 4 % (weight/vol) (Sigma-Aldrich. Cat. No.: P6148) in PBS, pre-warmed at 37 °C. Cells were then washed with PBS 3 times, 5 min each and non-specific binding sites were blocked with a solution of saponin (Sigma-Aldrich. Cat. No.: 47036) 0.05% (weight/vol)/bovine serum albumin (Sigma-Aldrich. Cat. No.: A7906) 0.2% (weight/vol) in PBS (blocking solution from now on) for 20 min at RT. Primary antibodies were diluted in blocking solution and cells were incubated with this solution overnight at 4 °C. A list of all primary antibodies used is shown in supplementary table 3. After primary antibody incubation, cells were washed 3 times with blocking solution, 5 min each. All secondary antibodies were diluted to 2 µg/µl in blocking solution. Cells were incubated with the secondary antibodies' solution for 1 h at RT. Cells' nuclei were stained with 4′, 6-diamidino-2-phenylindole, dihydro-chloride (DAPI) (Invitrogen. Cat. No.: D3571) at 0.5 µg/µl. DAPI was added to the secondary antibodies' solution. Finally, cells were washed 3 times with blocking solution, 3 times with PBS, rinsed in Milli-Q water once, and mounted on

a microscopy slide with FluorSave® reagent (CalBiochem. Cat. No.: 345789). When antibody-independent labelling of the plasma membrane was required, after the incubation with the secondary antibodies, cells were washed 3 times, 5 min each with PBS and incubated for 10 min at RT with a 1:200 dilution in PBS of the lipophilic dye CellBrite™ (Biotium. Cat. No.: 30023). Finally, cells were washed 3 times with PBS and mounted on a microscopy slide with FluorSave.

## Confocal microscopy

High resolution images were acquired in an inverted Leica TCS SP8 microscope equipped with 405, 458, 476, 488, 496, 514, 561, and 633 nm laser lines, photomultipliers (PMT) and hybrid detectors. Unless otherwise indicated, a plan apo 63X, 1.4 NA, oil-immersion objective was used for specimen imaging. Pixel size and z-step sizes were set to fulfil Nyquist criterion. In all cases, laser and detection spectral bands were chosen to maximize signal recovery while avoiding signal bleed-through. Scanning speed was set to 600 Hz and bidirectional. Between 1 and 3 lines were averaged to generate the final image. When sequential scanning was needed to avoid signal bleed-through, each acquisition sequence was done in between lines. Bit-depth was set to 8 bits.

## Live cell imaging

Image acquisition was done in an Andor Revolution XD Spinning Disk confocal microscope equipped with 405, 445, 488, 514, and 561 nm laser lines. Digitalization of the fluorescent emission was done with an EM CCD camera of $512 \times 512$ pixels and $0.37594\,\mu m$ xy resolution. Electron multiplier gain was set to 300. Exposure time was set to 300 ms and bit-depth was 14 bits. Imaging medium was 1.8 ml of isotonic buffer (see secretion assays of materials and methods for composition). One complete z-stack acquisition was done every 20 s for 5 min before ATP addition. After 5 min, 0.2 ml of 100 mM ATP was added to the cells to reach a final ATP concentration of $100\,\mu M$. Cells were imaged every 30 s for another 25 min.

## Forster resonance energy transfer (FRET) measurement

For the energy donor and acceptor pair we selected NeonGreen and mScarlet respectively as they have been previously described as a suitable FRET pair[57]. We generated expression plasmids of the FRET pair linked to syntaxin-2 and Tspan-8 respectively. To measure the energy transfer between the two we used the "acceptor photobleaching" method for its simplicity and robustness as has been previously described[58–60]. Upon excitation of the donor, a fraction of the excitation energy is not emitted by donor fluorescence, but instead it's transferred to the acceptor that is in close proximity (donor quenching by the acceptor). This leads to a decrease in the donor emission intensity. In acceptor photobleaching, we compare the quenched and unquenched emissions of the donor by acquiring an image of the quenched donor (neonGreen·Stx2). We then photo-bleach the acceptor (Tspan-8·mScarlet) and acquire a second image of the unquenched donor. By comparing the change in intensity of the quenched (pre photobleaching) and unquenched (post photobleaching) images of the donor we can measure FRET between Stx2 and Tspan-8 using the following formula:

$$\text{FRET efficiency} = 1 - (\text{quenched donor}/\text{unquenched donor})$$

To calculate the FRET efficiency we used averaged intensity values of each image. To calculate FRET maps we used pixel by pixel calculation of the FRET efficiency. Prior to calculation, images were smoothed with a Gaussian filter (ratio 0.5 pixels), background-subtracted and converted to 32-bit images. All image calculations were done in Fiji/ImageJ.

## Dot and western blot quantification

ImageJ / Fiji[61,62] were used for the analysis of all digitalized images from dot and western blots experiments. To quantify ECL or fluorescence intensity, the background was subtracted from the images. We generated an intensity profile for the membrane and measured the area of the peaks of interest. These area values are a direct measure of the emission intensity of the bands of interest in the membrane. Only images with no saturated pixels were used for quantification.

## Co-localization quantification

For co-localization analysis we calculated the Pearson Correlation Coefficient (PCC) of confocal images of cultured HT29-N2 cells using Fiji/ImageJ[61,62]. In monocultures, we calculated PCC with the coloc2 plugin included in Fiji/ImageJ. To calculate the PCC for each optical slide of confocal images of the Caco2/HT29-N2 co-cultures we wrote an ImageJ macro to subtract the image's background and then calculate the PCC of the entire optical slide.

## Quantification of the percentage of secreted mucin granules in live cell imaging experiments

Quantification of spots (mucin granules) was done as previously described[53]. Spinning disk confocal images were analysed with the "spot detector"[63] plug-in in Icy software[64]. Confocal images were background-subtracted and aligned. To determine the number of spots (mucin granules) at each time point (frame), a wavelet adaptive threshold was computed using a combination of scales 2 and 3 of the spot detector plug-in. Size thresholds were set between 30 and 80. With these settings, the spot-detector plug-in computes the number of spots for each optical plane at each frame of the time-lapse. The results of the spot detector were saved in a spreadsheet table. With the statistical software R, we calculated the number of mucin granules at each frame as the sum of the detected spots in each of the optical slides. To express the change of the amount of mucin granules relative to initial amount (before ATP addition), the number of spots of each frame were divided by the average number of spots of the frames before ATP addition (15 frames; 5 min). These values give us the number of granules in cells at each time point. The inverse value is the number of granules that are released. To express these values as a percentage we multiplied the proportion of secreted granules by 100.

## Statistical analysis

All statistical analysis and graphical representation of the data was performed with the R software, several packages from the tidyverse[65] and the data.table package. When the experimental design required a single pairwise comparison, Student t tests were applied. When the experimental design required the comparison of multiple conditions with a single control condition an ANOVA with orthogonal contrasts was applied. In the cases where we needed to do multiple comparisons, we applied an ANOVA test followed by a Tukey's HSD test. When the experimental design contained more than one principal factor, we used two-way ANOVA with interaction. The exact value of n, what it represents and centre and dispersion measures are described in the figures and figure legends. The exact p value of the statistical test is written in the figures or figure legends. Shapiro-wilk and Levene tests were done to control for normal distribution of and equal variance of the samples.

## Statistics and reproducibility

Unless otherwise stated, secretion assays were performed three times by triplicates each time. Each triplicate is an independent culture and independently treated. The triplicates were processed in parallel. In the statistical model these triplicates are nested within the main factor, typically genotype or treatment.

Figure 4: One of three independent co-immuno precipitation experiments are shown in figures a to f.

Figure 5: Images were obtained from three independent co-cultures. One representative image is shown in figures a, b, d and e. All images taken were included in the quantification of the PCC and showed similar results.

Figure 6b, e, g, i, k and m: The red dot is the mean value of the gray dots +/− the standard deviation.

Figure 6i, k and m: Grouped in different colors are samples that were processed in parallel.

## Reporting summary

Further information on research design is available in the Nature Portfolio Reporting Summary linked to this article.

## Data availability

Source data are provided with this paper. In the source data file, raw quantification data and annotated raw images of the blots shown in this manuscript and are provided in a spreadsheet. The raw data of each figure is placed in a tab with the corresponding figure name. The raw data generated in this study has been deposited in a Zenodo repository under accession code 7680682[66] and is publicly available under Creative Commons Attribution 4.0 International license. The data for the bioinformatics analysis has been previously reported by Deprez et al.[32] and is publicly available through an interactive web tool [https://www.genomique.eu/cellbrowser/HCA/]. Source data are provided with this paper.

## Code availability

The R code to reproduce the bioinformatics analysis of the data published by Deprez et al.[32] is publicly available on GitHub [https://zenodo.org/record/7954818][67] under the GNU General Public license version 3.

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

## Acknowledgements
We thank all members of the Malhotra laboratory, especially Ishier Raote, and Meir Aridor from the University of Pittsburgh School of Medicine for valuable discussions. Cell sorting experiments were carried out by the joint CRG/UPF FACS Unit at Parc de Recerca Biomèdica de Barcelona (PRBB). Fluorescence microscopy was performed at the Advanced Light Microscopy Unit at the CRG, Barcelona. We acknowledge the work of the Free Software Foundation [https://www.gnu.org/] in general and in particular to the JabRef Development Team (2021) for providing and promoting the use of software libre. JabRef [https://www.jabref.org] - An open-source, cross-platform citation and reference management software. We acknowledge the support of the Spanish Ministry of Science and Innovation to the EMBL partnership, the Centro de Excelencia Severo Ochoa, and the CERCA Programme / Generalitat de Catalunya. V.M. is an Institució Catalana de Recerca i Estudis Avançats professor at the Centre for Genomic Regulation. Work in the Malhotra lab is funded by grants from the Spanish Ministry of Economy and Competitiveness (Plan Nacional to V.M.: PID2019-105518GB-I00) and the European Union's HORIZON under the grant agreements No. 894115 and No. 101062382. J.W. is funded by the European Research Council (H2020-MSCA-IF-2019-894115). A.L.L. is funded by the European Research Council (HORIZON-MSCA-2021-PF-101062382). This work reflects only the authors' views, and the EU Community is not liable for any use that may be made of the information contained therein.

## Author contributions
Conceptualization, J.W. and V.M.; Methodology, J.W., A.L.L., G.B., O.F., C.A., M.P.R., and N.B.; Investigation, J.W., A.L.L., G.B., and V.M.; Data analysis, J.W.; Writing—Original draft, J.W. and V.M.; Funding acquisition, V.M., A.L.L. and J.W.; Supervision, J.W. and V.M.

## Competing interests
The authors declare no competing interests.
