## [Peer review file · Nature Communications]

REVIEWER COMMENTS

Reviewer #1 (Remarks to the Author):

This work by Wojnacki and colleagues explores the role of TSPAN8 in regulating fusion of mucin filled vesicles during the long phase of stimulated secretion in secretory cells of the epithelium. The authors identify a negative regulatory function of TSPAN8 on this process. In addition, they observed the formation of a complex between syntaxin 2 and 3 and TSPAN8 and suggest that this interaction is impeding interaction of syntaxin 2 with VAMP8, thus reducing docking and fusion of vesicles. Although the work is interesting and the experiments are of good quality, I have several concerns.

Major:

The main concern is regarding the mechanism hypothesised for TSPAN8 role. It is clear from the experiments that loss of TSPAN8 reduces significantly mucin secretion and that TSPAN8 can interact with syntaxin 2. However, the other experiments that try to identify a mechanism for this phenomenon are not convincing.

In particular, in Figure 4: TSPAN8 is precipitated with a very good efficiency; on the other hand, the levels of syntaxin 2 that are co-precipitated are very low. Considering VAMP8 would coprecipitate only through the interaction with syntaxin 2, I do not think that the absence of VAMP8 signal in this experiment is sufficient to claim a mutually exclusive interaction.

In my opinion, it would at least be necessary to repeat the experiment from the opposite side, immunoprecipitating VAMP8 and observing possible co-precipitation of syntaxin 2 (expected) and TSPAN8 (not expected). In addition to this, identification of the domains of TSPAN8 and syntaxin 2 could also strengthen this conclusion if the domain of syntaxin 2 interacting with TSPAN8 is the same that can interact with VAMP8.

Second, in Fig 6, that syntaxin 2 and TSPAN8 are in the same regulatory pathway for mucin vesicles secretion is not clearly demonstrated. The fact that reducing syntaxin 2 levels in the absence of TSPAN8 still reduces mucin secretion does not rule out whether the two proteins are in the same pathway or not. A better experiment would probably be to measure mucin secretion levels in cells KO for syntaxin 2 where TSPAN8 could be knocked down or knocked out. If TSPAN8 only exerts its function by blocking syntaxin 2, reducing TSPAN8 should not have any effect on mucin secretion of syntaxin 2 is absent.

Minor points:

Related to Figure 1 and the text related to it. I do not think it is correct to talk about "expression levels" of genes as readout of a single cell RNA sequencing. These experiments give reads of levels of RNAs which could be influenced also by RNA degradation or protection, not only by expression.

In Figure 2E, I do not understand what the "proportion of granules" is related to. It is unclear what and how it is measured, also in the methods section. This needs to be more clearly explained. In addition, the images seem very saturated, and I do not see how the authors could have been able to clearly separate individual vesicles.

Are syntaxin 2 levels at the membrane the same upon TSPAN8 knockout? This could be measured by applying antibodies directed towards the extracellular domain of syntaxin 2 to the cells before fixation (surface staining) or by biochemistry methods (surface biotinylation or BS3). Several tetraspanins have been associated with intracellular trafficking (Charrin et al., 2014), so it could also be that TSPAN8 regulates trafficking of syntaxin 2 to the plasma membrane. I believe this should be checked experimentally.

In fig 3A, the claim that TSPAN8 is localised at the plasma membrane only based on the colocalisation with Na⁺/K⁺-ATPase is not convincing, as the signal could also arise from intracellular localisation very

close to the plasma membrane. Either surface staining (as described in the point above) or biochemical methods (surface biotinylation or BS3 crosslinking) can be used to demonstrate this.

In the text related to figure 5A, it is stated that syntaxin2 does not localise to mucin granules. It is unclear how this would have been assessed as only colocalisation between syntaxin 2 and Na /K - ATPase is shown here. In addition, it looks to me that there some granules which are positive for both proteins in the lower Z plane image, so this is at least only partially true. More specific experiments, possibly checking colocalisation between mucin-GFP and syntaxin2, need to be performed here to claim this.

Fig 6A, that syntaxin 2 and TSPAN8 colocalise at the plasma membrane based simply on localisation of the two overexpressed proteins, seems excessive. Surface staining of the two proteins should be carried out here to demonstrate it. If antibodies for the endogenous proteins, that bind to the extracellular portion, are not available, overexpression of the two proteins carrying a tag which would be exposed extracellularly could be used as an alternative. At the very least, the authors should add the lipophilic marker in this figure and in that case tone down this result.

For dot blots images, always clearly state in the figure legend and in the figure itself, if what it has been measured is in lysate or media.

In blots with knockout cells, could the author please show also the knocked-out protein.

In fig 4A: From a quick search it would seem that the anti RFP antibody, used as control for the anti GFP precipitation, is of a different species (one rabbit and one mouse). This is not an acceptable control. Authors should either show IP with an IgG of the same species or another antibody of the same species as the anti GFP antibody.

The sentence "Tetraspanin transfection of Tspan-8-RFP...showed a very high degree of co-localization" needs to be rephrased.

Reviewer #2 (Remarks to the Author):

In this manuscript, the authors explored the role of TSPAN8 in the biphasic mucin secretion in human airway cell line. Previous work had shown that tetraspanin-8 contributes to tumour initiation, promotion and progression by acting as a signaling hub, while these authors here proposed interesting roles for TSPAN8 in regulating secretion of mucin granules. Although the study includes abundant experimental approaches and contains interesting observations, there are substantial problems in the work that raise questions about the major conclusions being made.

1. It appears that the entire study had used cell lines with KO and overexpression studies, and no studies in primary mucus-secreting goblet cells. There is a global Stx-2 (epimorphin) KO mouse used to study multiple cell types, including colonic cells, and therefore should mimic the mucin-secreting phenotype in this work. Gene deletion of the endogenous Stx2 in that mouse should have profound effects on mucus secretion in the airways and colon, based on the hypothesis of this work. These workers should really use that mouse model to strengthen their findings beyond this work on cell lines, otherwise their findings would have questionable physiologic relevance.

2. Figure 2. It would be good to have some EM studies performed to show the precise locations of the secretory granules and to define what are 'docked' and what are in the 'reserve pool'. Presumably, Stx2 KO or TSPAN8 overexpression (to sequester Stx2) would result in a clearing of granules from the docked sites on the apical PM especially after the initial stimulation to release the docked vesicles. This observation would better support the purported lack of recruitment of mucus granules to docking

sites.

3. Syntaxin-2 has a large intracellular domain followed by a hydrophobic transmembrane domain without extracellular domain, while tetraspanin-8 has been reported to have a very short intracellular amino- and carboxy-terminal tail (~7-9 amino acids) as well as a small inner loop (~4 amino acids) but has a large extracellular loop. It is difficult to envision how the long cytoplasmic domain of syntaxin-2 could be sequestered by such a short cytoplasmic domain of tetraspanin-8. A comprehensive investigation is required, such as mapping the precise domain of Tspan-8 that interacts with syntaxin-2 and what domain(s) within syntaxin-2.

4. Figure 4. The binding analysis of Tspan8 with Stx2 in this work with the impression it blocks binding to Munc18-1/2 and VAMP8 is incomplete. Changes in Stx configuration by Munc18 (and other priming protein, Munc13) enable Stx to form the complete SNARE complex with VAMPs and SNAP25(23). In this work, it appears that Stx2 does NOT require Munc18 but requires simply Tspan8 to release Stx2, and there was no full demonstration that this leads to full SNARE complex formation or how Munc18 was actually altered by Tspan8 to enable Munc18 ability to prime Stx2. There is lack of evidence of how precisely Tspan-8 interacts and activates Stx2. The strategy used was by overexpressing Stx2, then co-immunoprecipitating Tspan-8 from using HT29 cell lysates. The binding between Tspan-8 and Stx2 might not be by direct physical interaction of the two proteins but could potentially be confounded of the bait possible interacting with proteins that syntaxins are well known to bind to (Munc13, etc.). In vitro purified recombinant proteins to show direct interactions between Tspan8 and Stx2 should have been demonstrated. At the current sophisticated stage of SNARE membrane fusion biology, such detailed structure-functional data should have been performed. Syntaxins are known to be rather 'sticky', and Stx2 in particular has been reported to also stick to non-SNARE/Munc18 proteins.

5. "After 24 minutes TSPAN8 KO cells had secreted 10 % of the total amount of granules during the second phase of secretion." does not seem to be that strong an effect. I wonder if Stx3 (that Tspan8 also binds to) might have more effect on mucin secretion than Stx2. Although they focused on Tspan8 interaction with Stx2, it may be Tspan8 interaction with Stx3 that is also or more important. They should have done Stx3 deletion should to see how much it contributes to mucin secretion compared to Stx2 deletion and when both are deleted. And why would TSPAN8 bind/sequester Stx2 and not also sequester Stx3 that is so structurally similar to Stx2. While Stx2 seems co-localized with Tspan8 on the plasma membrane, where is Stx3 located the goblet cells? co-localized or not with Tspan8. The paper of Brunger they cited raises the importance of Stx3, rather than Stx2. Some recent reports seem to suggest the reduced profusion ability of Stx2, which should be cited and commented and compared to their results.

Reviewer #3 (Remarks to the Author):

In this manuscript, Wojnacki and colleagues through the use of microscopy, immunoprecipitation experiments and secretion assays describe a regulatory mechanism for the biphasic nature of induced secretory responses, which involves the sequestering of Stx-2 by Tspan-8 thereby limiting the docking of secretory granules to the plasma membrane. In general, I find this manuscript interesting however there are a number of important aspects of the data that I think should be improved to strengthen the conclusions drawn in this manuscript. My specific comments follow below:

1. The authors rely on the identification of Tspan-8 from a previously published single cell dataset of epithelial cells from the human airways. Data from this paper and other published datasets highlight the plasticity of mucus producing cells in the airways, with no clear single cell type responsible for the secretion of mucus. The study by Perez is unable to separate Club cells from 'true' goblet cells. I feel that the current manuscript would benefit from a more detailed analysis of goblet cells from other organs where the mucus producing cell population can be more clearly defined for example the gastrointestinal tract such as those from Haber et al (Nature 2017), Parikh et al (Nature 2019) and Nyström et al (Science 2021).

2. The authors extensively utilize HT29-N2 cells, a mucus producing subclone of HT29 cells. As this is a cell line originally isolated from a colon adenocarcinoma it is not clear what other unknown effects on cellular physiology have been acquired by these cells, for example they have a gastric mucin phenotype despite being a colonic cell line. Therefore, it is not entirely clear how closely this model reflects the in vivo situation. The manuscript would be greatly enhanced if these experiments were complemented by work in organoids/monolayers derived from healthy tissue, for example HBEC cultures if the focus of the paper is to be on the secretory responses in the airways or intestinal organoids/monolayers for the intestine.

3. The authors show the localization of Tspan-8 to the membrane in HT29-N2 cells; however, I am not aware of studies showing the localization of this protein in secretory cells in tissue samples. Immunostaining for Tspan-8 in tissue sections would provide important information not only for understanding the role of this protein during the suggested secretion process but also for the function of Tspan-8 in vivo.

4. In the proposed model of biphasic secretion, the balance of the Stx2/Tspan-8 and Stx2/VAMP8 is key to regulation of this process. These interactions are shown in Fig.4 in a series of immunoprecipitation experiments. However, these experiments do not show how the balance between these interactions changes during the switch from the rapid secretion phase to the slow-release phase. Showing how the levels of these interactions change through the time course of the experiment would provide stronger evidence to support the proposed model.

Minor Comments:

1. In the introduction it is stated that there are 5 gel forming mucins, this should be 4 gel forming mucins (MUC2, MUC5AC, MUC5B and MUC6).

2. Referring to HT29-N2 cells as HT29 cells can be confusing as the reader may mistake these cells for the non-mucus producing parental cell line.

3. The images supporting fig 2E are difficult to interpret, a series of images which cover the time course in more detail would be beneficial. Also, the inclusion of supporting time-lapse video of the imaging would help.

4. The inclusion of z-projections of confocal images used to show membrane localization of Stx2, Tspan-8 etc. would help the reader with interpretation of this data due to the polarized nature of differentiated epithelial cells.

5. The reason for changing the model to HT29-N2/Caco2 co-cultures is not explained.

6. Information on the anti-Muc5AC antibody used was not available on the supplier's website, this may have been a typo in the methods section. As different Muc5AC antibodies recognize mature or apoprotein information regarding the clone used would be useful.

7. Some of the data in supplemental information would be better placed in the main figures, for example Suppl fig 6 A-C could be moved to fig 5.

References Cited:

Haber et al 2017, Nature Vol. 551 Pages 333-339

Parikh et al 2019, Nature Vol. 567 Pages 49-55

Nyström et al 2021, Science Vol. 372 Pages eabb1590

General comments

1. A critical point is having better evidence for the Stx2/Tspan8 interaction, and considering the point made by Reviewer #2, identifying the required domains would be important.

We provide a solution to this extremely challenging concern. The best solution would be to purify the proteins, reconstitute their binding *in vitro*, obtain fragments to map the binding site, and ultimately the amino acids in the three dimensional structures. Or use Cryoelectron microscopy to identify the binding domains of the protein complex *in vitro* or in intact cells. But we hope the reviewer realizes the difficult task of purifying a tetraspanin. This would constitute a new project in itself and not something that can be addressed in a timely manner. Nevertheless, to gain some additional insights into the interaction between Stx2 and Tspan-8, we performed the following experiments and the data strengthening our proposal are included in the revised manuscript.

1a. We measured interaction between Stx2 and Tspan-8 by Forster Resonance Energy Transfer (FRET) in cells. For FRET to occur, the donor and acceptor molecules can not be separated by more than 9 angstroms. This distance between proteins can be considered to represent a direct interaction.

We generated two expression plasmids: neonGreen.Stx2 (energy donor) and Tspan-8.mScarlet (energy acceptor). These plasmids were used to create lentiviral particles. We coinfect HT29-N2 cells with these plasmids and selected cells stably co-expressing both proteins by Fluorescent Activated Cell Sorting (FACS).

To measure FRET between neonGreen.Stx2 (energy donor) and Tspan-8.mScarlet (energy acceptor), we used Acceptor Photobleaching for its simplicity and robustness. With this method, we acquired an image of the energy donor (neonGreen Stx2). We then photo-bleached the acceptor (Tspan-8 mScarlet) and acquired a second image of the donor in the same conditions as the first one. If in the pre bleaching image there was FRET between Stx2 and Tspan-8, part of the energy used to excite the neonGreen of Stx2 would have been transferred to mScarlet and would generate mScarlet emission. In the second image (post bleaching) there can't be any further transfer of energy from donor (neonGreen Stx2) to the acceptor (Tspan-8 mScarlet) as the acceptor is bleached. This way, in the second donor image the intensity would be higher compared to the first image if there was FRET between donor and acceptor as all the excitation energy would now be used for neonGreen emission and not mScarlet emission as in the pre bleaching donor image.

Representative images of the FRET energy donor pre and post bleaching are shown in Figure 4G of the revised manuscript. In these images, there is a clear increase in the fluorescence in the post bleached image. The images showing bleaching of the FRET acceptor are shown in Figure 4H and quantified in Figure 4J. The quantification shows near complete photo-bleaching of the acceptor (Tspan-8 mScarlet). As a technical control of the FRET calculation, we have acquired images pre and "post" bleaching, but for which the laser was kept at 0% during the "bleaching" part of the protocol.

To calculate the FRET efficiency we used the following formula:

$$\text{FRETeff} = 1 - (\text{Donor pre bleaching} / \text{Donor post bleaching})$$

The photo-bleaching is only of the FRET acceptor.

The quantification shows near 10% FRET efficiency between GFP Stx2 and Tspan-8 mScarlet (Figure 4K). The FRET map (Figure 4I) shows that FRET occurs in the region of the plasma membrane where both proteins are localized (arrows).

These data are shown in Figure 4 and the corresponding methodology is included in material and methods sections of the manuscript as are the appropriate references for this experimental procedures.

1b. We expressed Tspan-8 without the c-terminal residues (d234-237) in cells and found it to be retained in the ER. Interestingly, Stx2 is also arrested in the ER of such cells. This shows that these two proteins are in a complex and binding to Stx2 is independent of the C-terminal region of Tspan-8 (Figure 7 and see further explanation below).

Together, these two different approaches nicely support our proposal of a direct interaction between Tspan8 and Stx2.

2. The concern about potential artifacts due to the cancerous nature of the cell line that was studied is also significant.

The problem with the mucin secretion field is the difficulty in working with the goblet cells.

HT29-N2 cells are convenient for identifying components of the mucin secretion pathway because they can be differentiated into mucin producing cells by starvation for 7-10 days. The use Human Bronchial Epithelial Cells (HBEC) have the technical limitation that it takes 3 to 4 weeks to differentiate them into mucin producing cells. It is impossible to perform any experiments that utilize RNAi-based approaches to manipulate the endogenous levels of the target proteins. In addition, genetic modifications such as CRISPR or lentiviral expression, typically require a step of cell amplification, which is not possible with these cells as the number of division cycles are limited. Moreover, we would need to generate a stable line that expresses a tagged mucin, in this case MUC5AC or MUC5B, and we have been unable to select a line that stably expresses this tagged mucin in HBEC cells.

The other major problem is that it is unclear whether the available antibodies recognize all forms of a given mucin, which undergoes heavy glycosylation, condensation and then reopening of the condensed form in the extracellular space. Furthermore, the use of antibodies in mucin biology

makes it impossible to study the dynamic behavior of mucin secretion in live cells. The large size of gel-forming mucins makes transient expression of these proteins almost impossible.

We have managed to use CRISPR/Cas9 knockin approaches to finally select a line stably expressing MUC5AC-GFP at the endogenous level. This technical advance has made it possible to monitor the export of MUCAC by the live cell imaging.

In order to address the reviewers concerns, we have performed a different kind of experiment. As shown in the paper, loss of TSPAN8 increases mucin secretion. It is therefore reasonable to assume that over-expression of TSPAN8 would inhibit mucin secretion or secretion of another molecule that might utilize the same principle. We therefore chose to test the effect of TSPAN8 over expression on Insulin secretion. Insulin secretion is also mediated by large granules that undergo condensation. This experiment also has the potential of revealing whether TSPAN8 functions in secretion of molecules that are generally released by large granules. Insulin secretion requires syntaxin-1 and the respective cells are amenable to experimental manipulation. Mouse and rat, unlike the human, insulin-secreting beta cells do not express TSPAN8 (Champy et al. 2011), but it is likely that a related tetraspanin is involved. This is not surprising because our own data has shown that the colon cells express TRPM5 for mucin secretion, whereas the airway cells express TRPM4 for the same function (Our published work, Cantero-Recasens et al. 2019. JBC).

As Champy and colleagues found for mice beta cells, we could not detect TSPAN8 expression in rat insulin-secreting cells (INS-1) (Figure 8A). We cloned WT Tspan-8 c-terminally tagged with mScarlet into rat INS-1 cells by lentiviral infection. We selected cells expressing Tspan-8 mScarlet by FACS to generate a stable cell line. Confirmation for over expression was tested by Western Blot (Figure 8B) and confocal microscopy (Figure 8C).

Confocal microscopy of rat INS-1 cells showed that Tspan-8 mScarlet localized to plasma membrane and in some intracellular structures (Figure 8C; arrows and arrowheads respectively) as in HT29-N2 cells (Figure 3).

Immunoprecipitation of Tspan-8 mScarlet revealed the presence of syntaxin-1 in the precipitate (Figure 8 D). An insulin secretion assay in the presence of glucose in the culture media was used to measure insulin secretion by ELISA. This experiment showed that Tspan-8 over expression in INS-1 cells reduced the amount of glucose-stimulated insulin secretion (Figure 8E). These results confirm the generality of our proposed model that a Tspan-8 binds a plasma membrane specific syntaxin in a cell type specific manner to control granule release.

These data are shown in Figure 8 of the revised version of the manuscript.

- Finally, the experimental requests to better prove that Tspan8 and Stx2 are in the same regulatory pathway, and to show that there is a change in Stx2 interactions in the rapid-to-slow secretion shift, would greatly strengthen your manuscript.

To address this issue, we have used a CRISPR-modified cell line that expresses a truncated version of Tspan-8 in the c-terminus (d234-237). We show that the mutant Tspan-8 is retained at the endoplasmic reticulum and its location at the plasma membrane is reduced (Figure 7A and B). We also show that ER-trapped Tspan-8 d234-237 binds to Stx2 (Figure 7D) and retains Stx2 in the ER (Figure 7C). The amount of Stx2 at the plasma membrane is therefore reduced. This is a further test of the interaction of Tspan-8 with Stx2.

A mucin secretion assay shows that the cell line bearing the mutant Tspan-8 d234-237 secretes on average 50% less mucin5-AC compared to WT cells (Figure 7E and F).

These results show that:

- 1) The c-terminus of Tspan-8 is necessary for its export from the ER.
- 2) The c-terminus of Tspan-8 is dispensable for Stx2 interaction.
- 3) The relocation of Stx2 in Tspan-8 d234-237 KI cells supports our proposal on the interaction between the two proteins.
- 4) The defect in secretion in Tspan-8 d234-237 is due to reduction in the levels of Stx2 at the plasma membrane.

These data and the data shown in Figure 6 strongly suggest that Tspan-8 and Stx2 work in the same pathway and their interaction is necessary for controlling mucin secretion. The figure below showing the new data is included as a new figure (Figure 7) in the revised manuscript.

Our comments to the specific questions of the reviewers follow.

Reviewer #1 (Remarks to the Author):

This work by Wojnacki and colleagues explores the role of TSPAN8 in regulating fusion of mucin filled vesicles during the long phase of stimulated secretion in secretory cells of the epithelium. The authors identify a negative regulatory function of TSPAN8 on this process. In addition, they observed the formation of a complex between syntaxin 2 and 3 and TSPAN8 and suggest that this interaction is impeding interaction of syntaxin 2 with VAMP8, thus reducing docking and fusion of vesicles.

Although the work is interesting and the experiments are of good quality, I have several concerns.

Major:

The main concern is regarding the mechanism hypothesised for TSPAN8 role. It is clear from the experiments that loss of TSPAN8 reduces significantly mucin secretion and that TSPAN8 can interact with syntaxin 2. However, the other experiments that try to identify a mechanism for this phenomenon are not convincing.

In particular, in Figure 4: TSPAN8 is precipitated with a very good efficiency; on the other hand, the levels of syntaxin 2 that are co-precipitated are very low. Considering VAMP8 would coprecipitate only through the interaction with syntaxin 2, I do not think that the absence of VAMP8 signal in this experiment is sufficient to claim a mutually exclusive interaction.

In my opinion, it would at least be necessary to repeat the experiment from the opposite side, immunoprecipitating VAMP8 and observing possible co-precipitation of syntaxin 2 (expected) and TSPAN8 (not expected).

We have addressed the reviewers concern as suggested. We immunoprecipitated VAMP-8 and asked whether Stx2 or Tspan-8 are contained in the immunoprecipitate. We found that Stx2 co-precipitated with VAMP-8, but Tspan-8 was not detectable (Figure 4D). This result supports our hypothesis that binding of Tspan-8 to Stx2 makes the latter unable to engage with VAMP-8.

These data are included in Figure 4D of the revised manuscript.

In agreement with the remark of reviewer 2 in point 3 about the topology of these proteins, we agree that calling the interaction between Stx2 and Tspan-8 or VAMP-8 mutually exclusive might be excessive based on the current data. We have therefore changed the tone of our conclusion and state that when Tspan-8 binds Stx2, the latter is not able to engage in a fusion reaction, but Tspan-8 might not necessarily directly compete with the VAMP-8 binding site. The exact mechanism of this

interaction between Tspan-8 and syntaxin-2 and the exclusion of VAMP-8 from this complex awaits further experimentation.

In addition to this, identification of the domains of TSPAN8 and syntaxin 2 could also strengthen this conclusion if the domain of syntaxin 2 interacting with TSPAN8 is the same that can interact with VAMP8.

Tspan-8, as the name suggests, is mostly buried in the membrane. The N- and the C-termini are in the cytoplasm along with 4 loops. Syntaxin-2 is anchored to the plasma by its c-terminus and does not contain an extracellular domain. Mapping the exact binding site between these two proteins is not straightforward as explained above in the general statements to all reviewers. But based on the experiments, we can state with confidence that the C- terminus of Tspan-8 is not necessary for binding to Stx2 and explained further in more detail in below.

Second, in Fig 6, that syntaxin 2 and TSPAN8 are in the same regulatory pathway for mucin vesicles secretion is not clearly demonstrated. The fact that reducing syntaxin 2 levels in the absence of TSPAN8 still reduces mucin secretion does not rule out whether the two proteins are in the same pathway or not. A better experiment would probably be to measure mucin secretion levels in cells KO for syntaxin 2 where TSPAN8 could be knocked down or knocked out. If TSPAN8 only exerts its function by blocking syntaxin 2, reducing TSPAN8 should not have any effect on mucin secretion of syntaxin 2 is absent.

We have tried to knock down TSPAN8 in Stx2-depleted cells, but in our hands the KD was not effective. Based on experiments in which we have treated HT29-N2 cells with cycloheximide we suspect that Tspan-8 has a relatively long half life and therefore RNAi-based knockdowns do not work in our experimental set-up.

We have addressed this concern by using a mutant version of Tspan-8 that is retained in the ER and found that it sequesters Stx2 in the ER. This causes a decrease of Stx2 at the plasma membrane and explains the decrease in mucin5-AC secreted by cells expressing the mutant Tspan-8. This is shown in Figure 7 of the revised manuscript and explained in more detail in the comments to all reviewers.

These results show that Tspan-8 and Stx2 work together in the pathway of mucin secretion and also lend support to our proposal that their interaction is important for this function.

Minor points:

Related to Figure 1 and the text related to it. I do not think it is correct to talk about “expression levels” of genes as readout of a single cell RNA sequencing. These experiments give reads of levels of RNAs which could be influenced also by RNA degradation or protection, not only by expression.

We agree that the actual value obtained in scRNAseq analysis is likely a result of both expression and degradation. Nevertheless, we have decided to use the same nomenclature as the authors of the paper, Deprez et al., where they described the gene expression landscape of the human airway cells. In another recent scRNAseq study of the cell lineage hierarchies of human airway cells (Sandra Ruiz García et al. Development. 2019) they also use gene expression to refer to the amount of mRNA detected in single cells.

We would like to maintain the same nomenclature for better consistency with these earlier publications.

In Figure 2E, I do not understand what the “proportion of granules” is related to. It is unclear what and how it is measured, also in the methods section. This needs to be more clearly explained. In addition, the images seem very saturated, and I do not see how the authors could have been able to clearly separate individual vesicles.

The reviewers pointed out the Figure 2E of the original manuscript was difficult to interpret. We have thus modified the representation of our results on mucin secretion in live cells to make the data easier to follow.

As suggested by reviewer 3, we have changed the cells images for a series of images which cover the time course in more detail. We have selected a smaller area of a lateral projection of the time-lapse to make details more visible and marked the secretion events with arrows. In the image series we have included every other frame (separated by 40 seconds each) for space limitations.

We have also modified the graph showing quantification of mucin secretion to represent the proportion of granules that are secreted instead of the amount of granules inside cells.

Since the fluorescent intensity of the secreted mucin was much lower compared to the mucin granules, we intentionally saturated the images to make the secreted mucin more visible as this is the most important message of the experiment. The original images are not saturated. In the revised version of the manuscript we have included an unsaturated time-lapse video of mucin secretion.

The quantification of the number of mucin granules was done with the “spot detector” plug in of the Icy bioimaging software. For quantification, we used the original images that are not saturated and maintain the three-dimensional distribution of the granules. The spot detector plug in does not rely on intensity for granule detection, but uses an object-based approach instead.

We have extended and clarified the explanation of this experiment in the results and material and methods sections of the revised manuscript.

The updated version of Figure 2 where we show the results of the live-imaging experiments follows:

Are syntaxin 2 levels at the membrane the same upon TSPAN8 knockout? This could be measured by applying antibodies directed towards the extracellular domain of syntaxin 2 to the cells before fixation (surface staining) or by biochemistry methods (surface biotinylation or BS3). Several tetraspanins have been associated with intracellular trafficking (Charrin et al., 2014), so it could also be that TSPAN8 regulates trafficking of syntaxin 2 to the plasma membrane. I believe this should be checked experimentally.

Syntaxin-2 does not contain an extracellular domain

(https://www.uniprot.org/uniprotkb/P32856/entry#subcellular_location) that could be used for surface staining or biotinylation. We have tried to measure the amount of Stx2 at the plasma membrane in WT and TSPAN8 KO cells by purifying the plasma membrane fraction with the commercial kit from abcam ([ab65400](https://www.abcam.com/products/ab65400)). Although we could detect Stx2 at the plasma membrane-enriched fraction, the results were highly variable even within same samples, so we could not reliably use these data for quantification.

Nevertheless, the detection of Stx2 at the plasma membrane of WT and TSPAN8 KO HT29-N2 cells and the quantification of the Pearson Correlation Coefficient with the plasma-membrane localized protein NaKATPase alpha1 (Figure 5) strongly suggest that trafficking of Stx2 to plasma membrane is unaltered in TSPAN8 KO cells. This makes another important point that Tspan-8 is not some specialized chaperone for Stx2 trafficking because without Tspan-8, Stx2 still reaches the cell surface. But when a mutant Tspan-8 (d234-237) is expressed in cells and retained in the ER, it also causes retention of Stx2. Therefore, wherever Tspan-8 is localized, it binds/interacts and holds Stx2.

In fig 3A, the claim that TSPAN8 is localised at the plasma membrane only based on the colocalisation with Na /K -ATPase is not convincing, as the signal could also arise from

intracellular localisation very close to the plasma membrane. Either surface staining (as described in the point above) or biochemical methods (surface biotinylation or BS3 crosslinking) can be used to demonstrate this.

In Figure 3C of the manuscript, we show co-localization of Tspan8 GFP with a lipophilic marker membrite (<https://biotium.com/product/membrite-fix-cell-surface-staining-kits/>). This dye first labels the plasma membrane and only after 30 min translocates to intracellular structures. The imaging of live cells expressing Tspan-8 GFP and stained with Membrite was performed within 30 minutes after addition of the dye. Furthermore, we have never seen intracellular structures labeled with membrite during imaging. These results, in addition to the colocalization shown in Figure 3 reveal the plasma membrane localization of Tspan-8.

We have nevertheless, extended our observations by purifying a plasma membrane fraction (ab65400) of WT HT29-N2 cells and immunoblotting Tspan-8. These results confirm the enrichment of Tspan-8 at the plasma membrane fraction. We have included these new results in Figure 3D.

In the text related to figure 5A, it is stated that syntaxin2 does not localise to mucin granules. It is unclear how this would have been assessed as only colocalisation between syntaxin 2 and Na /K - ATPase is shown here.

In the first version of the manuscript, the image showing syntaxin-2 labeling and mucin5AC GFP was in Figure Supplementary 6C whereas quantification of the Pearson Correlation Coefficient (PCC) between the two was in Figure 5B for WT cells and Figure Supplementary 6B for TSPAN8 KO cells. In the image of Figure Supplementary 6C, we can see Stx2 localized to the plasma membrane while the vast majority of mucin5AC is localized in cytosolic granules, consistently with the very low PCC shown in Figure 5B (violet line and dots).

To make these data more clear, and as suggested by reviewer 3 in minor comments 7, we have moved Figures Supplementary 6A, B and C of the original manuscript to main Figure 5D, E and F in the revised version. We have also included images of syntaxin-2 labeling in TSPAN8 KO cells expressing mucin5-AC GFP. These images and the quantification of the Pearson Correlation Coefficient between Stx2 and mucin5-AC GFP show the lack of co-localization between the two.

The images below show the signal of Stx2 and mucin5-AC in basal (lower row) and apical (upper row) regions of HT29-N2 cells in co-culture with Caco2 cells. Left column are WT cells while right column are TSPAN8 KO cells. These images are now included in the main Figure 5B and E.

In addition, it looks to me that there some granules which are positive for both proteins in the lower Z plane image, so this is at least only partially true. More specific experiments, possibly checking colocalisation between mucin-GFP and syntaxin2, need to be performed here to claim this.

Specific analysis of Stx2 and mucin5-AC co-localization is shown in Figure 5B and E and quantified in Figure 5C and F (purple line and dots) in the revised manuscript. This analysis shows virtually no colocalization between the two. The marginal increase in PCC towards the apical region is a reflection of the poor quality of the anti Stx2 antibody, which shows considerable level of non specificity. The low but visible signal of the antibody staining of Stx2 observed in the cell nucleus shows that there is some degree of non-specific labeling by this antibody.

The Stx2-positive, dot-like signal visible in the basal regions of goblet cells in Figure 5A are not mucin granules and they only show marginal overlap with the NaK-ATPase signal. Mucin granules concentrate in the apical part of goblet cells as shown in Figure 5B (WT) and E (TSPAN8 KO) while the mentioned co-localization is in the basal part of the cell where no mucin is detected.

Fig 6A, that syntaxin 2 and TSPAN8 colocalise at the plasma membrane based simply on localisation of the two overexpressed proteins, seems excessive. Surface staining of the two proteins should be carried out here to demonstrate it. If antibodies for the endogenous proteins, that bind to the extracellular portion, are not available, overexpression of the two proteins carrying a tag which would be exposed extracellularly could be used as an alternative. At the very least, the authors should add the lipophilic marker in this figure and in that case tone down this result.

Syntaxin-2 does not contain an extracellular region that could be used for surface staining. We have also tested the anti Tspan-8 antibody (ab70007) for surface staining, but this did not work in our hands.

We have however, stained endogenous Stx2 by immunolabeling and labeled the plasma membrane with the membrite lipophilic marker used in other experiments. Although fixation disrupts to some degree the distribution and intensity of Tspan-8 GFP (CRISPR modified cell line) and the efficacy of the anti syntaxin-2 antibody was not optimal, we detected Tspan-8GFP and Stx2 at the plasma membrane (arrows). We have included these images in Figure 6C of the revised manuscript.

We consider that the data presented in figure 6A, B and C, the plasma membrane localization of Tspan-8 described in Figure 3, the plasma membrane-like distribution of Stx2 described in Figure 5 and the interaction between Stx2 and Tspan-8 at the plasma membrane described by FRET in Figure 4 of the revised manuscript adequately support our conclusion that Stx2 and Tspan-8 co-localize and interact at the plasma membrane.

For dot blots images, always clearly state in the figure legend and in the figure itself, if what it has been measured is in lysate or media.

We have included this information in the legends and figures now.

In blots with knockout cells, could the author please show also the knocked-out protein.

We have included these images in Figure 2 E and F and Figure 5H.

In fig 4A: From a quick search it would seem that the anti RFP antibody, used as control for the anti GFP precipitation, is of a different species (one rabbit and one mouse). This is not an acceptable control. Authors should either show IP with an IgG of the same species or another antibody of the same species as the anti GFP antibody.

The intended purpose of the antiRFP-coated beads was to control the potential non-specific binding of proteins to the beads or even antibodies themselves. We have used antiRFP- (5F8) and antiGFP-coated (3H9 alpha) beads because they come from the same company (chromotek and now called protein tech) and the agarose beads are similar in both products. Both, anti-RFP and anti-GFP antibodies on these beads are nanobodies derived from alpaca camelid and are therefore from the same species. We forgot to include the reference number of the antiRFP-coated control beads in our previous version of the manuscript. We have now included this information in the revised manuscript and clarified the use of the antiRFP-coated beads as controls for non-specific binding of proteins from the input. This information is included in the “Immunoprecipitation” section of the materials and methods.

In Figure 4B to E and Figure Supplementary 5 there are additional control conditions that confirm the specific coIP of Stx2 when we immuno-precipitate Tspan-8.

The sentence “Tetraspanin transfection of Tspan-8-RFP...showed a very high degree of co-localization” needs to be rephrased.

We could not locate this specific sentence the reviewer is referring to. Perhaps he/she meant the “Transient transfection of Tspan-8-RFP, immunofluorescence of ...” which is just below the title that begins with “Tetraspanin-8 is localized...” Tetraspanin (from the title) and Transient (from the paragraph) are very close physically and they are similarly spelled, this may have caused the confusion.

Reviewer #2 (Remarks to the Author):

In this manuscript, the authors explored the role of TSPAN8 in the biphasic mucin secretion in human airway cell line. Previous work had shown that tetraspanin-8 contributes to tumour initiation, promotion and progression by acting as a signaling hub, while these authors here proposed interesting roles for TSPAN8 in regulating secretion of mucin granules. Although the study includes abundant experimental approaches and contains interesting observations, there are substantial problems in the work that raise questions about the major conclusions being made.

1. It appears that the entire study had used cell lines with KO and overexpression studies, and no studies in primary mucus-secreting goblet cells. There is a global Stx-2 (epimorphin) KO mouse used to study multiple cell types, including colonic cells, and therefore should mimic the mucin-secreting phenotype in this work. Gene deletion of the endogenous Stx2 in that mouse should have profound effects on mucus secretion in the airways and colon, based on the hypothesis of this work. These workers should really use that mouse model to strengthen their findings beyond this work on cell lines, otherwise their findings would have questionable physiologic relevance.

Our work goes beyond cell lines in two different ways:

1) Our bio-informatic analysis of human lung cells. This analysis shows a specific increase in the expression of TSPAN8 in mucin-secreting human cells. The functional reason for this increase before our work was not known.

2) The insulin secretion assay data with INS-1 cells is now included in the revised version of the manuscript and described in more detail in the general comments to the reviewers.

Working with human primary cells is both a technical challenge and ethically complex as is working with animal models. In our work we aim to bridge this challenge by combining a bio-informatic analysis of human lung cells, and use a biological system that allows us to decipher the mechanism of action of a protein with unknown function.

Our interest in Stx2 goes along with its role in secretion as a binding partner of Tspan-8. The function of syntaxins by binding a vesicle associated membrane protein (VAMP) has already been described, therefore we consider that the proposed experiments in an animal KO for STX2 would not add much to our proposed model.

Our principal aim is to understand the cellular mechanism of protein secretion. The detection of a relevant gene (TSPAN8) for protein secretion by bio-informatics and the subsequent work with cell lines would allow us to rapidly advance the understanding of mucin secretion in a field that has not seen much development in terms of the genes involved in controlling quantity and quality of mucins secreted by cells. The new experimental data in the revised manuscript and the data presented in the first version of our work, strongly support our conclusions and provide a plausible explanation for the increased expression of TSPAN8 in mucin-secreting cells.

2. Figure 2. It would be good to have some EM studies performed to show the precise locations of the secretory granules and to define what are 'docked' and what are in the 'reserve pool'. Presumably, Stx2 KO or TSPAN8 overexpression (to sequester Stx2) would result in a clearing of

granules from the docked sites on the apical PM especially after the initial stimulation to release the docked vesicles. This observation would better support the purported lack of recruitment of mucus granules to docking sites.

We do not have access to an EM facility and this would require setting up a collaboration with another lab. But it is important to remember that given the enormous relative size of mucin granules, to precisely determine the minimal distance separating them from the plasma membrane would require a comprehensive and detailed 3D reconstruction of at least the apical part of the cell. Also, the granules are not in uniform size. Immature granules change their size before they are fully mature and able to dock and fuse with the plasma membrane. This increases the level of complexity of the analysis. The reviewers make an interesting point, but this is beyond the scope of this manuscript.

3. Syntaxin-2 has a large intracellular domain followed by a hydrophobic transmembrane domain without extracellular domain, while tetraspanin-8 has been reported to have a very short intracellular amino- and carboxy-terminal tail (~7-9 amino acids) as well as a small inner loop (~4 amino acids) but has a large extracellular loop. It is difficult to envision how the long cytoplasmic domain of syntaxin-2 could be sequestered by such a short cytoplasmic domain of tetraspanin-8. A comprehensive investigation is required, such as mapping the precise domain of Tspan-8 that interacts with syntaxin-2 and what domain(s) within syntaxin-2.

We agree that the lack of a large intracellular domain in Tspan-8 makes it difficult to envision a direct competition with VAMP-8 for a binding site in Stx2, somehow similar to what has been proposed for tomosyn (Fujita et al. *Neuron*. 1998; Sauvola et al. *Elife* 2021). However, and as described in our comments to reviewer 1, we propose that binding of Tspan-8 to Stx2 could keep Stx2 in a microdomain of the plasma membrane that is physically separated from other SNAREs and related proteins, without directly interacting through its SNARE domain. In support of this model, the large extracellular domain of tetraspanins has been proposed to facilitate the formation of tetraspanin-enriched microdomains and that the short intracellular N- and C-termini can facilitate protein-protein interactions (Charrin et al. *J Cell Sci*. 2014; Termini and Gillette. *Front Cell Dev Biol*. 2017). We propose that Tspan-8 could be enriched in, and drive the formation of microdomains in the plasma membrane. The binding of Tspan-8 with Stx2 (data from coIP and FRET analysis) could serve to drag and sequester Stx2 into these domains where it would be unable to bind other SNAREs.

4. Figure 4. The binding analysis of Tspan8 with Stx2 in this work with the impression it blocks binding to Munc18-1/2 and VAMP8 is incomplete. Changes in Stx configuration by Munc18 (and other priming protein, Munc13) enable Stx to form the complete SNARE complex with VAMPs and SNAP25(23). In this work, it appears that Stx2 does NOT require Munc18 but requires simply Tspan8 to release Stx2, and there was no full demonstration that this leads to full SNARE complex formation or how Munc18 was actually altered by Tspan8 to enable Munc18 ability to prime Stx2. There is lack of evidence of how precisely Tspan-8 interacts and activates Stx2. The strategy used was by overexpressing Stx2, then co-immunoprecipitating Tspan-8 from using HT29 cell lysates. The binding between Tspan-8 and Stx2 might not be by direct physical interaction of the two proteins but could potentially be confounded of the bait possible interacting with proteins that syntaxins are well known to bind to (Munc13, etc.). In vitro purified recombinant proteins to show direct interactions between Tspan8 and Stx2 should have been demonstrated. At the current

sophisticated stage of SNARE membrane fusion biology, such detailed structure-functional data should have been performed. Syntaxins are known to be rather 'sticky', and Stx2 in particular has been reported to also stick to non-SNARE/Munc18 proteins.

We appreciate the thoughtful comments of the reviewer on our proposed model. The sophisticated understanding of SNAREs function is the result of 3 decades of work from a huge number of labs. We do not pretend to achieve the same level of understanding of Tspan-8 beyond what we have presented here in one paper. In fact, the earlier models of SNAREs mode of interaction were different from the recent models and even in such a highly studied system there are many concerns and lot still remain to be known. Our data adds a new component to the complexities of the mechanism regulating membrane fusion. Our proposed model does not negate the involvement of Munc proteins in the cycle of SNARE proteins during SNARE-mediated granule fusion. We propose that Tspan-8 acts by keeping Stx2 altogether outside of the described cycle of activation, fusion, and recycling mediated by Munc13, Munc18, Vamp, NSF and others. Obviously, there are still aspects of our proposed model that we do not fully understand and we hope that our data will bring others into addressing the function of Tspan-8 and other tetraspanins in events controlling SNARE-mediated fusion.

5. "After 24 minutes TSPAN8 KO cells had secreted 10 % of the total amount of granules during the second phase of secretion." does not seem to be that strong an effect. I wonder if Stx3 (that Tspan8 also binds to) might have more effect on mucin secretion than Stx2. Although they focused on Tspan8 interaction with Stx2, it may be Tspan8 interaction with Stx3 that is also or more important. They should have done Stx3 deletion should to see how much it contributes to mucin secretion compared to Stx2 deletion and when both are deleted. And why would TSPAN8 bind/sequester Stx2 and not also sequester Stx3 that is so structurally similar to Stx2. While Stx2 seems co-localized with Tspan8 on the plasma membrane, where is Stx3 located the goblet cells? colocalized or not with Tspan8. The paper of Brunger they cited raises the importance of Stx3, rather than Stx2. Some recent reports seem to suggest the reduced profusion ability of Stx2, which should be cited and commented and compared to their results.

We agree with the reviewer that Stx3 might also be required or involved in mucin secretion. Furthermore, our experimental data shows that Stx3 also co-immuno-precipitates with Tspan-8 so if Stx3 was to play a role in mucin secretion it might be subjected to the same regulatory process imposed by Tspan-8. The fact that Tspan-8 over expression affects Stx1 mediated insulin secretion (new data in the paper – Figure 8) suggests that a tetraspanin and syntaxin complex works in a cell type dependent manner. The isoforms likely vary, but the overall mechanism is the same.

Our data shown in Figure 5G – J of the revised manuscript shows that in STX2 KO cells there is still some residual mucin secretion. This points to the involvement of another syntaxin in the mucin granule fusion. Brunger and colleagues recently showed in an *in vitro* system that Stx3 can mediate vesicle fusion provided some accessory proteins are present, but syntaxins *in vitro* can be relatively promiscuous for its binding partners (Yang, B. et al. J Biol Chem. 1999; Fasshauer et al. J Biol Chem. 1999). The data presented by Brunger and colleagues suggests that Stx3 can potentially function for mucin secretion but this remains to be tested in cells. Our data also supports this hypothesis and predicts that if stx3 does participate in mucin secretion then it will be subjected to the regulation by Tspan-8 along with Stx2.

To extend our observations of Stx2 to Stx3 would not add much to the general mechanism of action of Tspan-8 we are currently proposing.

Reviewer #3 (Remarks to the Author):

In this manuscript, Wojnacki and colleagues through the use of microscopy, immunoprecipitation experiments and secretion assays describe a regulatory mechanism for the biphasic nature of induced secretory responses, which involves the sequestering of Stx-2 by Tspan-8 thereby limiting the docking of secretory granules to the plasma membrane. In general, I find this manuscript interesting however there are a number of important aspects of the data that I think should be improved to strengthen the conclusions drawn in this manuscript. My specific comments follow below:

1. The authors rely on the identification of Tspan-8 from a previously published single cell dataset of epithelial cells from the human airways. Data from this paper and other published datasets highlight the plasticity of mucus producing cells in the airways, with no clear single cell type responsible for the secretion of mucus. The study by Perez is unable to separate Club cells from 'true' goblet cells. I feel that the current manuscript would benefit from a more detailed analysis of goblet cells from other organs where the mucus producing cell population can be more clearly defined for example the gastrointestinal tract such as those from Haber et al (Nature 2017), Parikh et al (Nature 2019) and Nyström et al (Science 2021).

Our bioinformatician is working with these datasets, but we have encountered some setbacks. We will gladly include anything we learn in the paper once we manage to complete the analysis. It is worth noting that this was not an absolute requirement but a suggestion to help learn more about this protein in colon cells. We appreciate the suggestion and will comply once we learn anything new from our efforts.

2. The authors extensively utilize HT29-N2 cells, a mucus producing subclone of HT29 cells. As this is a cell line originally isolated from a colon adenocarcinoma it is not clear what other unknown effects on cellular physiology have been acquired by these cells, for example they have a gastric mucin phenotype despite being a colonic cell line. Therefore, it is not entirely clear how closely this model reflects the in vivo situation. The manuscript would be greatly enhanced if these experiments were complemented by work in organoids/monolayers derived from healthy tissue, for example HBEC cultures if the focus of the paper is to be on the secretory responses in the airways or intestinal organoids/monolayers for the intestine.

We have included new experimental data in the revised manuscript and described above in detail to general comments of the reviewers. We also explain the reason for the systems that we have used in our work in point 1 of reviewer 1, which is further stated below:

The problem with the mucin secretion field is the difficulty in working with the goblet cells.

HT29-N2 cells are convenient for identifying components of the mucin secretion pathway because they can be differentiated into mucin producing cells by starvation for 7-10 days. The use Human Bronchial Epithelial Cells (HBEC) have the technical limitation that it takes 3 to 4 weeks to differentiate them into mucin producing cells. It is impossible to perform any experiments that utilize RNAi-based approaches to manipulate the endogenous levels of the target proteins. In addition, genetic modifications such as CRISPR or lentiviral expression, typically require a step of cell amplification, which is not possible with these cells as the number of division cycles are

limited. Moreover, we would need to generate a stable line that expresses a tagged mucin, in this case MUC5AC or MUC5B, and we have been unable to select a line that stably expresses this tagged mucin in HBEC cells.

The other major problem is that it is unclear whether the available antibodies recognize all forms of a given mucin, which undergoes heavy glycosylation, condensation and then reopening of the condensed form in the extracellular space. Furthermore, the use of antibodies in mucin biology makes it impossible to study the dynamic behavior of mucin secretion in live cells. The large size of gel-forming mucins makes transient expression of these proteins almost impossible.

We have managed to use CRISPR/Cas9 knockin approaches to finally select a line stably expressing MUC5AC-GFP at the endogenous level. This technical advance has made it possible to monitor the export of MUC5AC by the live cell imaging.

But to address the reviewers concerns, we have performed a different kind of experiment. As shown in the paper, loss of TSPAN8 increases mucin secretion. It is therefore reasonable to assume that over-expression of TSPAN8 would inhibit mucin secretion or secretion of another molecule that might utilize the same principle. We therefore chose to test the effect of TSPAN8 over expression on Insulin secretion. Insulin secretion is also mediated by large granules that undergo condensation. This experiment also has the potential of revealing whether TSPAN8 functions in secretion of molecules that are generally released by the large granules. Insulin secretion requires syntaxin-1 and the respective cells are amenable to experimental manipulation. Mouse and rat, unlike the human, insulin-secreting beta cells do not express TSPAN8 (Champy et al. 2011), but it is likely that a related tetraspanin is involved. This is not surprising because our own data has shown that the colon cells express TRPM5 for mucin secretion, whereas the airway cells express TRPM4 for the same function (Our published work, Cantero-Recasens et al. 2019. JBC).

As Champy and colleagues found for mice beta cells, we could not detect TSPAN8 expression in rat insulin-secreting cells (INS-1) (Figure 8A). We cloned WT Tspan-8 c-terminally tagged with mScarlet into rat INS-1 cells by lentiviral infection. We selected cells expressing Tspan-8 mScarlet by FACS to generate a stable cell line. Confirmation for over expression was tested by Western Blot (Figure 8B) and confocal microscopy (Figure 8C).

Confocal microscopy of rat INS-1 cells showed that Tspan-8 mScarlet localized to plasma membrane and in some intracellular structures (Figure 8C; arrows and arrowheads respectively) as in HT29-N2 cells (Figure 3).

Immunoprecipitation of Tspan-8 mScarlet revealed the presence of syntaxin-1 in the precipitate (Figure 8 D). An insulin secretion assay in the presence of glucose in the culture media was used to measure insulin secretion by ELISA. This experiment showed that Tspan-8 over expression in INS-1 cells reduced the amount of glucose-stimulated insulin secretion (Figure 8E). These results confirm the generality of our proposed model that a Tspan-8 binds a plasma membrane specific syntaxin in a cell type specific manner to control granule release.

These data are shown in Figure 8 of the revised version of the manuscript.

3. The authors show the localization of Tspan-8 to the membrane in HT29-N2 cells; however, I am not aware of studies showing the localization of this protein in secretory cells in tissue samples. Immunostaining for Tspan-8 in tissue sections would provide important information not only for understanding the role of this protein during the suggested secretion process but also for the function of Tspan-8 in vivo.

Unfortunately, we could not access human tissue samples during the time of the review of this manuscript. The issues are with the administrative clearance required to work with human samples and location of such samples for our work. This is a huge challenge that would require no less than a year to complete and beyond our abilities at present.

Perhaps there is some relief from this concern by our new data on the involvement of Tspan-8 in glucose mediated insulin secretion. Though technically challenging, we infected INS-1 rat cells with a fluorescent version of Tspan-8 and by live confocal imaging determined that in INS-1 rat cells Tspan-8 localizes very similar, if not identical, to its location in HT29-N2 cells (Figure 8C of the revised version of the manuscript). This confirms that Tspan-8 contains in its structure the necessary information for its localization at the plasma membrane regardless of the cell type.

4. In the proposed model of biphasic secretion, the balance of the Stx2/Tspan-8 and Stx2/VAMP8 is key to regulation of this process. These interactions are shown in Fig.4 in a series of immunoprecipitation experiments. However, these experiments do not show how the balance between these interactions changes during the switch from the rapid secretion phase to the slow-release phase. Showing how the levels of these interactions change through the time course of the experiment would provide stronger evidence to support the proposed model.

We immunoprecipitated Tspan-8 before and after ATP stimulation and determined the amount of Stx2 in the co-precipitated material. We did not detect any difference under these conditions but this most likely reflects the lack of quantitative nature of co-immunoprecipitation assays.

We also tried to determine by live FRET between Tspan-8 and Stx2 whether their binding was affected during mucin secretion. Although the FRET pair Tspan-8 mScarlet and GFP Stx2 worked and allowed us to determine their direct interaction by a non-biochemical method, the FRET efficiency between this FRET pair was too low for fast live cell imaging.

To improve the FRET pair to properly assess the change in their binding is beyond our methodological capacities. Even without these data, we consider the principal message of our work is unchanged and valid.

Minor Comments:

1. In the introduction it is stated that there are 5 gel forming mucins, this should be 4 gel forming mucins (MUC2, MUC5AC, MUC5B and MUC6).

MUC19 has also been described as a secreted, gel-forming mucin expressed in corneal epithelial cells, conjunctival goblet and epithelial cells, lacrimal gland cells, mucous cells of the submandibular gland, submucosal gland of the trachea and by middle ear epithelial cells.

As this is the most recent gel-forming mucin to be described we have added to our manuscript the specific references where this has been described in more detail:

Yu et al. 2008. MUC19 expression in human ocular surface and lacrimal gland and its alteration in Sjögren syndrome patients.

Chen et al. 2004. Genome-wide search and identification of a novel gel-forming mucin MUC19/Muc19 in glandular tissues.

2. Referring to HT29-N2 cells as HT29 cells can be confusing as the reader may mistake these cells for the non-mucus producing parental cell line.

We agree with this comment and we have changed HT29 to HT29-N2 in our manuscript.

3. The images supporting fig 2E are difficult to interpret, a series of images which cover the time course in more detail would be beneficial. Also, the inclusion of supporting time-lapse video of the imaging would help.

The time-lapse videos of this experiment were submitted as supplementary data. We are sorry that this reviewer did not receive these data.

As suggested by the reviewer we have changed the cells images for a series of images which cover the time course in more detail. We have selected a smaller area of a lateral projection of the time-lapse to make details more visible and marked the secretion events with arrows. In the image series we have included every other frame (separated by 40 seconds each) for space limitations.

We have also modified the graph showing the quantification of mucin secretion to represent the proportion of granules that are secreted instead of the amount of granules inside cells.

We have also explained in more detail how to interpret images in figure 2 of our manuscript and as suggested by reviewers 1 and 2, we have changed representation of some of the images in figure 2 to make them easier to understand.

The section of the material and methods has also been updated to better explain the quantification of the live-imaging experiments.

We repeat here the updated Figure 2.

4. The inclusion of z-projections of confocal images used to show membrane localization of Stx2, Tspan-8 etc. would help the reader with interpretation of this data due to the polarized nature of differentiated epithelial cells.

We have included 3D projection images of the most relevant confocal images in the manuscript as supplementary movie files.

5. The reason for changing the model to HT29-N2/Caco2 co-cultures is not explained.

HT29-N2 polarization is much more effective and pronounced under the co-culture conditions. We used this system to better visualize the distribution of Stx2 in the basal and the apical region of the cells. We have changed the text to better explain the use of this co-culture system.

6. Information on the anti-Muc5AC antibody used was not available on the supplier's website, this may have been a typo in the methods section. As different Muc5AC antibodies recognize mature or apoprotein information regarding the clone used would be useful.

We apologize for the inclusion of a wrong catalog number. We have corrected this in the revised manuscript.

The anti mucin5-AC antibody is the clone 45M1 manufactured by Labvision Neomarkers, catalog number: MS-145-P0. This antibody is now sold by thermofisher scientific, but that specific catalog product has been discontinued.

7. Some of the data in supplemental information would be better placed in the main figures, for example Suppl fig 6 A-C could be moved to fig 5.

We agree with the reviewer's remark and we have moved supp figure 6A to C to the main figure 5D to F.

References Cited:

Haber et al 2017, Nature Vol. 551 Pages 333-339

Parikh et al 2019, Nature Vol. 567 Pages 49-55

Nyström et al 2021, Science Vol. 372 Pages eabb1590

REVIEWER COMMENTS

Reviewer #1 (Remarks to the Author):

The authors answered my questions

Reviewer #2 (Remarks to the Author):

These workers have only partly addressed some of my previous concerns, specifically the fourth and fifth of my previous 5 comments, with some additional biochemical and clever FRET analysis of Tspan8/Stx2 binding interactions. Overall, they have now convinced me that Tspan8 acts mainly to: 1) sequester Stx2 from binding most (or all) of its partners and therefore should block subsequent sequential steps of docking, priming and fusion events: and 2) this sequestering effect of Tspan8 is not specific to Stx2 but would also apply to Stx3 and Stx1 (in INS-1) using overexpression models. Even with the new data they have generated, they have not addressed my previous first three more important comments.

While initial work they've done so far has suggested the above hypothesis, this was not unequivocally demonstrated to my satisfaction – that the mucus secretory granules would NOT dock onto docking sites in the context of a mucus/goblet cell– this was my previous Comment 2. While it is difficult to envision how exactly Tspan8 peculiar structure of short N- and C-termini could sequester these Stxs presumably by binding to as yet undefined Stx2 cytoplasmic domain, they've now instead proposed that Tspan8 could cluster into plasma membrane microdomains to do this – since this is the main point of their hypothesis, this has to be demonstrated, perhaps by TIRF microscopy – this would then satisfactorily address my previous Comment 3.

The inability to fully address my previous comments and those of the other two reviewers seems mainly due to the inherent limitations of the mucus cell line models they've used and their inability to use primary cell models, which they have reiterated in their rebuttal – this explanation is insufficient. It seems that have been trying to justify their Tspan8 – Stx2 model based on their initial bioinformatic analysis. The field of SNARE proteins biology has gone beyond simply demonstrating novel protein-protein interactions using overexpression systems – all such observations have to be unequivocally demonstrated in vivo to prove that they can be translated to genuine physiology beyond the current 'cell line' phenomenology. If Stx2 or Tspan8 is/are really so critical for mucus secretion, then an in vivo Stx2 KO or Tspan8 KO model should show various clinical phenotypes caused by the deficient mucus secretion from the lung, colon, etc, which could then be further verified with in vitro cell models - this would address my previous Comment 1. The Stx2/epimorphin KO mouse is in fact available from Jackson Laboratory (strain RRID:IMSR JAX:034237).

Reviewer #3 (Remarks to the Author):

The manuscript Wojnacki et al has been improved in revision. The presentation of the data, in particular the secretion data is easier to follow and interpret. The addition of the live FRET experiments strongly points towards an interaction between Tspan-8 and Stx2 (but does not discount a possible role for Stx3). It is unfortunate that the experiment did not allow for fast live cell imaging, but I understand the difficulties in performing such an experiment.

In my previous review I raised concerns regarding the nature of the cell model used in this study. The inclusion of data from the insulin secretion model is of interest and points to a possible more general function of tetraspanins in regulating biphasic secretion events, it does not completely address my concerns about the use of a cancer cell line as a model for mucus secretion. As goblet cells exist and

function in conjunction with other cell types *in vivo*, and both mucus secreting and ion transporting cells contribute to both mucus properties and secretory responses *in vivo*, I feel that other primary culture models would strengthen the manuscript. I feel that at a minimum the shortcomings of the model used in this study must be addressed in detail in the discussion, but preferably the inclusion of data from a primary cell model would be strongly encouraged.

Reviewer #2.

'The authors' have now instead proposed that Tspan8 could cluster into plasma membrane microdomains to do this – since this is the main point of their hypothesis, this has to be demonstrated, perhaps by TIRF microscopy – this would then satisfactorily address my previous Comment 3.

We are not changing our initial proposal. The direct physical interaction between stx2 and tspan8 has to happen to inhibit stx2 from engaging in a fusion event. What we claim is that the interaction doesn't need to be with the SNARE domain of stx2. It could be that tspan8 forms micro-domains (as it has been suggested for tetraspanins - <https://www.nature.com/articles/srep12201>) at the plasma membrane and the interaction with stx2 could sequester the later in these domains preventing its participation in a fusion event.

This has to be demonstrated, perhaps by TIRF microscopy.

This is good for small vesicles fusing to a large surface area. Goblet cells are not designed for this purpose and even a bigger challenge is that mucin granules are on the apical side of the cells while TIRF is typically performed on the basal side. Regarding mouse KO studies. This is simply not possible because we haven't the license to use animals and especially to expose them to agonists to stimulate mucin production and secretion. We also do not have an assay to measure the different phases of the bi-phasic release. Moreover, there is no reason to believe that the mouse release mechanism is similar to that of humans. This is clearly something that we plan to pursue with organoids derived from human cells in the future.

Reviewer #3, states that we should list the caveats of our approach to clarify its potential limitations. You will notice that we have changed the abstract, the second section of the results and the last part of the discussion clearly stating the limitations of our approach vis a vis physiological relevance.